# MiR-146a wild-type 3′ sequence identity is dispensable for proper innate immune function in vivo

Grant Bertolet[1], Natee Kongchan[2], Rebekah Miller[6], Ravi K Patel[6], Antrix Jain[3], Jong Min Choi[3], Alexander B Saltzman[4], Amber Christenson[2], Sung Yun Jung[3,4], Anna Malovannaya[3,4,5], Andrew Grimson[6], Joel R Neilson[1,2,5]

The prevailing model of microRNA function is that the "seed region" (nt 2–8) is sufficient to mediate target recognition and repression. However, numerous recent studies have challenged this model, either by demonstrating extensive 3′ pairing between physically defined miRNA–mRNA pairs or by showing in *Caenorhabditis elegans* that disrupted 3′ pairing can result in impaired function in vivo. To test the importance of miRNA 3′ pairing in a mammalian system in vivo, we engineered a mutant murine *mir-146a* allele in which the 5′ half of the mature microRNA retains its wild-type sequence, but the 3′ half's sequence has been altered to robustly disrupt predicted pairing to this latter region. Mice homozygous or hemizygous for this mutant allele are phenotypically indistinguishable from wild-type controls and do not recapitulate any of the immunopathology previously described for *mir-146a*–null mice. Our results indicate that 3′ pairing is dispensable for the established myeloid function of this key mammalian microRNA.

## Introduction

miRNAs are well-established regulators of mRNA stability and translation (Bartel, 2018). The discovery that hundreds of distinct miRNAs are expressed in animals (Lagos-Quintana et al, 2001; Lau et al, 2001; Lee & Ambros, 2001) and the subsequent finding that, unlike plant miRNAs, bilaterian animal miRNAs were only rarely predicted to base pair to their regulatory targets with full complementarity (Mansfield et al, 2004) raised the challenge of predictively identifying miRNA targets via computational analysis. Seminal work by the Burge and Bartel laboratories revealed that the most highly conserved pairing between miRNAs and mRNA 3′ UTRs was predicted to occur between nt 2 and 8 of mammalian miRNAs and their mRNA targets (Lewis et al, 2003). This finding was broadly incorporated into a generation of miRNA target prediction algorithms, and several subsequent studies corroborated these initial predictions by showing that the interaction between previously established miRNA–mRNA pairs was not only dependent upon the seed sequence but, moreover, independent of or only marginally reliant on non-seed sequence (Doench & Sharp, 2004; Kloosterman et al, 2004; Brennecke et al, 2005; Lai et al, 2005; Lim et al, 2005).

The first unbiased identification of bona fide miRNA/mRNA target pairs by physical association of Argonaute proteins with mRNA (Chi et al, 2009) corroborated the predictions of these existing algorithms to some extent by showing that most identified targets contained a seed match, a result which would be echoed by subsequent studies using similar methodologies (Hafner et al, 2010; Chi et al, 2012; Loeb et al, 2012; Grosswendt et al, 2014; Moore et al, 2015; Broughton et al, 2016). Some of these latter studies would also establish that target sites that contained a seed were typically more effective than sites that did not (Chi et al, 2012; Loeb et al, 2012; Moore et al, 2015). In addition to this, however, these studies also showed that the vast majority of these Argonaute/mRNA interactions also incorporated binding beyond the seed, within the 3′ region, and a significant proportion of microRNA response elements (MREs) contained no identifiable seed match at all. This growing body of literature has lent support to the notion that various degrees of base pairing between miRNA 3′ sequences and associated mRNA targets is the rule, rather than the exception. A functional role for 3′ pairing has also recently been corroborated in vitro in mammalian cells (Jeong et al, 2017) and in vivo in *Caenorhabditis elegans* (Zhang et al, 2015; Broughton et al, 2016; Brancati & Großhans, 2018). The collective implication of these studies is that the 3′ region of a given miRNA may confer an important but not-yet-fully defined role in determining the target spectrum (and thus function) of the miRNA. However, this has not previously been genetically addressed in vivo in a mammalian system.

We thus constructed a "flipped" allele of an miRNA whose loss-of-function has been previously established to yield a robust

---

[1]Graduate Program in Immunology, Baylor College of Medicine, Houston, TX, USA   [2]Department of Molecular Physiology and Biophysics, Baylor College of Medicine, Houston, TX, USA   [3]Mass Spectrometry Proteomics Core, Baylor College of Medicine, Houston, TX, USA   [4]Verna and Marrs McLean Department of Biochemistry and Molecular Biology, Baylor College of Medicine, Houston, TX, USA   [5]Dan L. Duncan Comprehensive Cancer Center, Baylor College of Medicine, Houston, TX, USA   [6]Department of Molecular Biology and Genetics, Cornell University, Ithaca, NY, USA

Correspondence: neilson@bcm.edu

phenotype in mice. Within this allele, the sequence of the 3′ half of the mature miRNA was exchanged with that of its complementary strand sequence in the pre-miRNA hairpin. We chose for our model, miR-146a, a pivotal immunoregulatory miRNA. MiR-146a is one of the two members of the *mir-146* family, the other being miR-146b, which differs from miR-146a in its mature sequence by only two nucleotides in the 3′ region. Despite this similarity, the two miRNAs are not functionally redundant *in vivo*; mice deficient for *mir-146a* but retaining wild-type *mir-146b* function are hypersensitive to LPS challenge and greatly predisposed to the development of hyper-immunity and myelodysplasia (Boldin et al, 2011). Given this robust phenotype, we bred mice homozygous for the "3′ flip (3′F)" allele to test how disruption of the 3′ region of this particular miRNA might impact *miR-146a*'s established genetic function. Closely following the workflow initially described for the characterization of *mir-146a*–deficient mice (and using the line as a control), we directly compared the phenotype of the *mir-146a$^{3'F}$*, *mir-146a*-null, and wild-type alleles, both in the context of homozygosity and heterozygosity.

## Results

### MiR-146a 3′ pairing contributes to the regulation of target UTRs *in vitro*

We first used *in vitro* reporter assays to determine whether, in this context, miRNA "sensors" responded to ectopic miR-146a expression in a manner similar to UTRs that had previously been established to be miR-146a targets. To this end, we designed two synthetic miRNA duplexes: one identical to the duplex produced after cleavage of the wild-type *mmu-mir-146a* pre-miRNA by Dicer and the other designed after the theoretical 3′F duplex (Fig S1A). The mutant sequence was modeled to reflect the mature strand of a pre-miRNA in which each nucleotide from positions 13 to 20 of the 5′ strand was exchanged with the opposite nucleotide from the 3′ strand and vice versa. We reasoned that this mutation would be the most deleterious to any potential physiological pairing by this region of the mature miRNA by making the 3′ region of the mature miRNA as near to its complement as possible without disrupting the secondary structure of the hairpin. Positions 9–12 within the mature miR-146a were left unperturbed under the rationale that these sequences were likely to be unpaired in a physiological setting (Schirle et al, 2014; Bartel, 2018), and positions 21–22 were conserved to lessen the chance that changes to loop-adjacent sequences might impact the processing or regulation of the resulting mutant.

We constructed luciferase reporter constructs containing six tandem MREs specifically tailored to match either of these mimics within the *pRL-TK-CX6X* backbone (Doench et al, 2003). Thus, both the WT and 3′F mimics would perfectly base pair to their cognate reporter (with the exception of nt 10–13 to eliminate "slicer" activity), but only base pair to their non-cognate reporter via nt 1–9 (Fig S1B). We transfected HeLa cells with these reporters, challenging each reporter separately with a series of concentrations of both the wild-type and mutant siRNA mimic duplexes. In both cases, transfection of the mimic designed to pair to the reporter via

both its seed region and the 3′ region repressed reporter activity to a significantly greater degree than the mimic designed to pair only via the seed region (Fig S1C and D, *P*-value < 0.0001, two-way ANOVA). These data demonstrated that extensive 3′ pairing has the potential to confer additional efficacy to miRNA function. However, because conventionally designed miRNA reporters, such as those assessed here, permit (or preclude) 3′ pairing to a more extreme extent than is thought to generally occur in a physiological setting, we next asked whether a similar phenomenon might be observed using reporters containing the 3′ UTRs of established miR-146a targets.

We thus repeated the reporter assay, this time targeting the 3′ UTRs of the two originally characterized targets of miR-146a, *Irak1*, and *Traf6* (Taganov et al, 2006). Alignment of the 3′F miR-146a sequence with the known MREs in both targets predicted substantial disruption of 3′ base pairing (Fig S2A and B). In keeping with this, we observed a modest but statistically significant (*P* < 0.0001, two-way ANOVA) reduction in the ability of the 3′F mimic to repress both the *Irak1* and *Traf6* 3′ UTRs (Fig S2C and D), whereas neither allele was capable of repressing mutated versions of the 3′ UTRs in which the central four seed nucleotides of each MRE had been mutated to their complement (Fig S2E and F). Taken together, these results support the notion that the 3′ non-seed region of miR-146a positively contributes to target repression *in vitro*.

### Construction and validation of a murine *mir-146a$^{3'F}$* allele

To determine whether the engineered "flip" mutation would be compatible with processing and expression from endogenous context, we synthesized and then subcloned the wild-type and 3′F miR-146a precursor hairpin sequences into the 3′ UTR of the *pRL-TK-CX6X* expression vector. Transfection of this vector into HEK 293 cells revealed processing and production of pre-miRNA and mature miRNA species from the mutant construct at absolute levels essentially identical to those observed for a similarly engineered wild-type miR-146a precursor construct in this context (Fig 1A and data not shown). We, therefore, targeted the endogenous *mir-146a* allele in murine embryonic stem cells via homologous recombination to replace the wild-type *mmu-mir-146a* locus with the mutant 3′F sequence (Fig 1B). In parallel, we deleted the *mir-146a* sequence in a second embryonic stem cell line, generating a null allele to serve as a positive control for *miR-146a*'s established phenotype (Fig 1C–F and data not shown). Mice derived from these embryonic stem cell lines and homozygous for either mutant allele were born at Mendelian frequency with no discernible overt phenotypes.

To confirm that the *mir-146a$^{3'F}$* allele was expressed at similar levels to wild-type *mir-146a*, we compared the relative levels of mature wild-type and 3′F miR-146a in several tissues and immune lineages derived from homozygous animals using qRT-PCR. Absolute values for miRNA expression in each sample were calculated via comparison with serial dilutions of synthetic standards cognate to each predicted mature miRNA sequence, and cross-priming analysis revealed a minimum of ~200-fold specificity of each primer set for its respective target (Fig 1G). Absolute expression of the mature miR-146a$^{3'F}$ miRNA in *mir-146a$^{3'F/3'F}$* homozygous mice

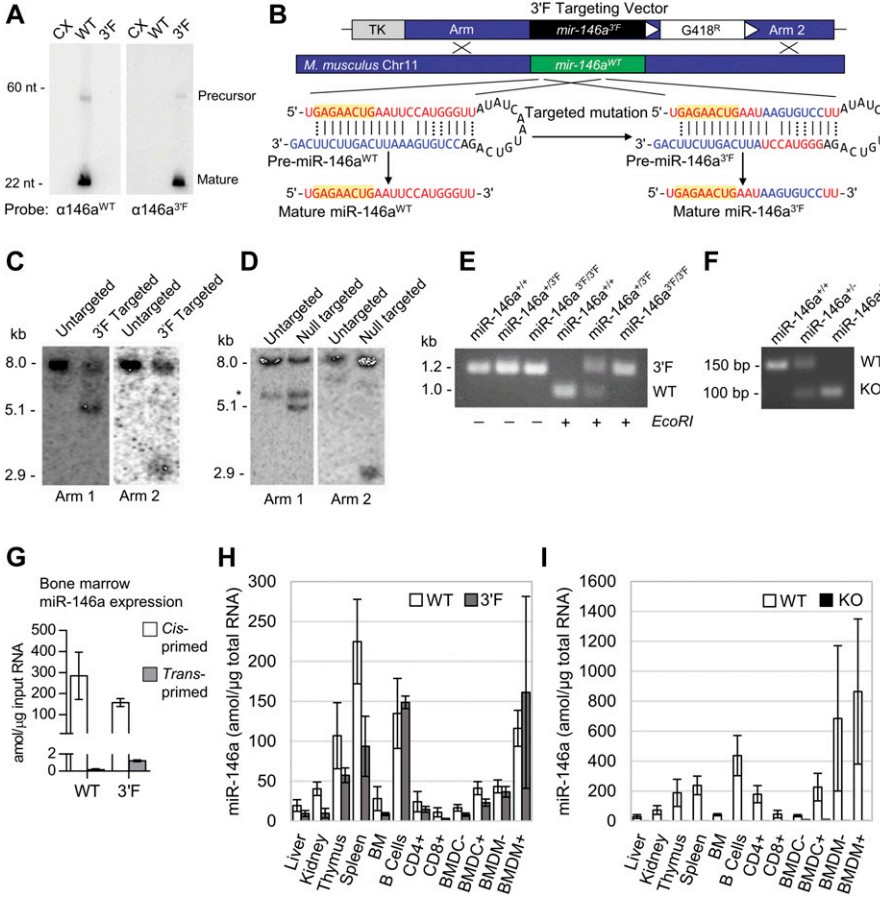

**Figure 1. Construction of *mir-146a* 3′F and KO mice.**
**(A)** Northern analysis of miR-146a[3′F] and miR-146a[WT] maturation from synthetic constructs transiently transfected into HEK 293T cells. 30 μg of 293 RNA transfected with each construct was loaded into the indicated lanes. CX = CXCR4 vector. **(B)** Schematic of mutation and gene targeting strategy. **(C)** Southern analysis of targeted *mir-146a[3′F]* locus in murine ESCs. **(D)** Southern analysis of targeted *mir-146a* KO locus in murine ESCs *cross-hybridizing band. **(E)** PCR genotyping of founder *mir-146a[3′F]* mice. The 3′F mutation disrupts an *EcoR1* site within the wild-type *mir-146a* sequence. **(F)** PCR genotyping of *miR-146a[−/−]* founder mice. **(G)** Cross-priming analysis to determine the specificity of miR-146a[WT] and miR-146a[3′F] primer sets. **(H)** Direct qRT-PCR quantitation (with standard) of both WT and 3′F miR-146a in indicated organs from *mir-146a[+/+]* and *mir-146a[3′F/3′F]* mice. **(I)** Direct qRT-PCR quantitation (with standard) of WT miR-146a in indicated organs from *mir-146a[+/+]* and *mir-146a[−/−]* mice. Error bars = SD. In (H) and (I), CD4[+] = CD4[+] T cells; CD8[+] = CD8[+] T cells; BMDCs and BMDMs were either stimulated (+) for 24 h before lysis with LPS or else unstimulated (−).

was similar to absolute expression of the wild-type *mir-146a* allele in homozygous wild-type mice in most tissues and cell populations examined (Fig 1H). As expected, neither wild-type nor 3′F miR-146a was detected in any tissues or cell populations derived from *mir-146a[−/−]* mice (Fig 1I and data not shown). Consistent with the qRT-PCR data, small RNA sequencing from BMDMs (Table S1) confirmed that mature miR-146a[3′F] was processed at the correct register within cells derived from *mir-146a[3′F/3′F]* mice (Fig S3A and B). Interestingly, these data also suggested higher levels of mature miR-146b within cells derived from *mir-146a[−/−]* mice, perhaps reflecting a pressure to compensate for the lack of miR-146a (Fig S3C). However, this increase in expression was not observed in the BMDMs of *mir-146a[3′F/3′F]* mice.

### Comparison of lifespan, hematology, and autoimmunity

We next set out to characterize mice homozygous for the *mir-146a[3′F]* allele in comparison with *mir-146a-null* and wild-type animals, closely adhering to the workflow described in the initial published analysis of *mir-146a* deficiency (Boldin et al, 2011). As previously described, *miR-146a[−/−]* mice were marked by decreased long-term viability when compared with wild-type littermates, beginning at 6 to 8 mo of age (Fig 2A). Gross analysis of *mir-146a[−/−]* mice sacrificed during this period revealed marked splenomegaly (Fig 2B and C)

and several signs of autoimmunity. Splenic T lymphocytes were characterized by increased basal activation on both CD4[+] and CD8[+] T lymphocytes, as defined by increased surface expression of CD69 and CD44 and decreased surface expression of CD62L. Spleens derived from *mir-146a[−/−]* mice were also characterized by an increase in GR-1[+]CD11b[+] myeloid compartment size (Fig 2E and F), and the sera of these mice were marked by an elevated titer of anti–double-stranded DNA antibodies (Fig 2G). Also consistent with previous work, aged *mir-146a[−/−]* mice (~14 mo) were characterized by marked decreases in hematocrit, red blood cell count, and platelet (Fig 2H). Although we did not observe previously described reductions in the representation of white blood cells or hemoglobin at 14 mo of age (Boldin et al, 2011), our *mir-146a[−/−]* line exhibited significant increases in mean corpuscular volume, mean platelet volume, hemoglobin distribution width, and red cell distribution width (Fig S4), consistent with anemia and bone marrow failure. Histological analysis of the liver and kidney of aged mice from our *mir-146a[−/−]* line revealed increased infiltration of mononuclear inflammatory cells in both organs as compared with wild-type controls (Fig 2J and K). However, in contrast to what we had observed in vitro, none of these phenotypes, whether previously described or novel to this study, were observed in age-matched *mir-146a[3′F/3′F]* mice. Indeed, *mir-146a[3′F/3′F]* mice were phenotypically indistinguishable from wild-type controls (Fig 2A–K), whether

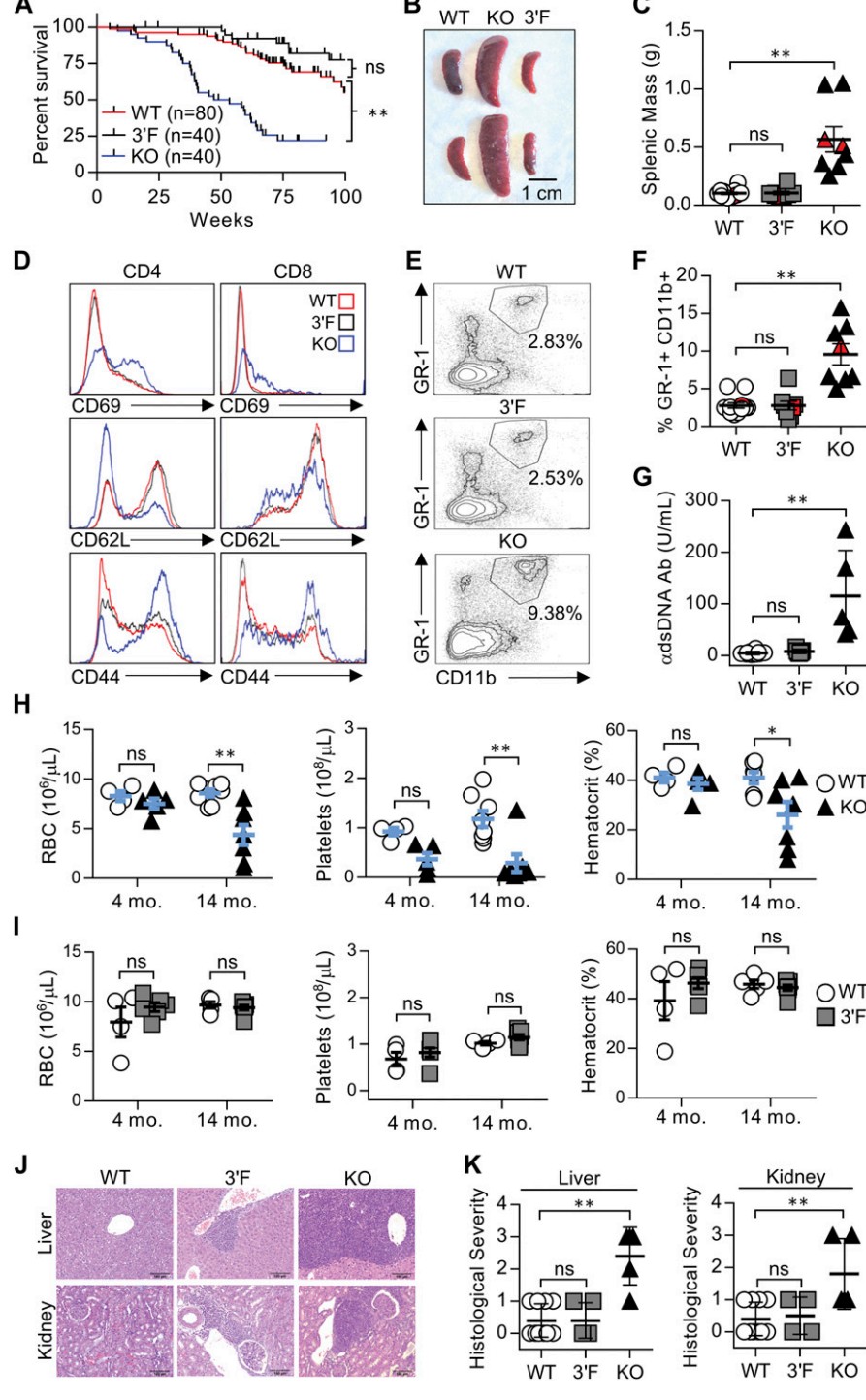

**Figure 2. The 3′F allele does not recapitulate the systemic hyperimmune phenotypes associated with *mir-146a* deficiency.**

**(A)** Kaplan–Meier plot of WT, KO, and 3′F mice viability. Mantel–Cox log-rank test. **(B)** Representative image of spleens harvested from mice of indicated genotype. **(C)** Quantitation of harvested splenic mass from mice of indicated genotypes. n = 12 (WT), 8 (3′F), and 8 (KO). Red data points correspond to the spleens pictured. Spleens taken from mice aged 225 ± 8 d. **(D)** Representative histograms of flow cytometry data assessing CD69, CD62L, and CD44 expression on splenic T cells. Average age: 240 ± 25 d. **(E)** Representative flow cytometry plots illustrating the relative size of the splenic GR-1+CD11b+ myeloid compartment in animals of indicated genotype. **(F)** Quantitation the splenic GR-1+CD11b+ myeloid compartment in animals of indicated genotype. Red data points correspond to spleens pictured. n = 12 (WT), 8 (3′F), and 8 (KO). Average age: 240 ± 25 d. **(G)** ELISA quantification of serum anti-dsDNA immunoglobulin. n = 12 (WT), 7 (3′F), and 5 (KO). Average age: 522 ± 27 d. Error bars = SD. **(H)** Comparative quantitation of indicated CBC parameters for WT versus KO mice. One-way ANOVA with Holm–Sidak post hoc correction for multiple comparisons. 4-mo average age 134 ± 11 d; n = 4 (WT) and 5 (KO). 14-mo average age 445 ± 25 d; n = 8 (WT) and 7 (KO). **(I)** Comparative quantitation of indicated CBC parameters for WT versus 3′F mice. One-way ANOVA with Holm–Sidak post hoc correction for multiple comparisons. 4-mo average age: 129 ± 10 d; n = 4 (WT) and 6 (3F). 14-mo average age: 442 ± 24 d; n = 5 (WT) and 7 (3′F). **(J)** Representative H&E–stained histological sections illustrating degree of leukocytic infiltration of the liver and kidneys within the mice of designated genotype. Organs harvested from mice aged 240 ± 25 d. **(K)** Comparative quantitation of histopathological index scores from different individuals. n = 10 (WT); 5 liver, 4 kidney (3′F); and 5 (KO). Error bars = SD. Statistical comparisons were done using unpaired, two-tailed *t* test unless otherwise indicated. *$P < 0.05$; **$P < 0.01$ in all subpanels. Error bars = SEM unless otherwise noted.

at the previously described time points or when assessed again in the context of extremely advanced age (~125 wk, or roughly 2.2 yr, Fig 3).

To better rule out the possibility that the *mir-146a*$^{3'F}$ allele was modestly hypomorphic, we repeated this battery of experiments, this time comparing *mir-146a*$^{3'F/−}$ hemizygotic mice with *miR-146a*$^{+/−}$ controls. Once again, no statistically sufficient differences

in phenotype were observed when these two lines were compared at the established time points, and no difference in longevity between the lines was observed out to 100 wk (Fig 4). Thus, for the phenotypes assessed, our data show that the *mir-146a*$^{3'F}$ allele is functionally indistinguishable from the wild-type *mir-146a* allele.

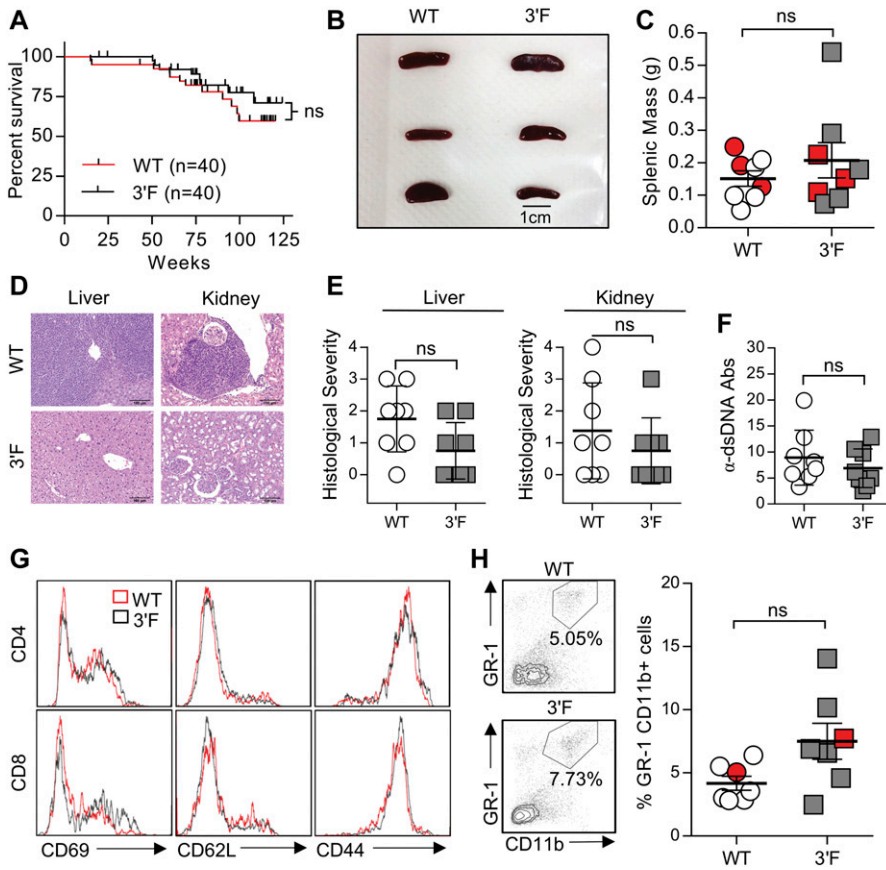

**Figure 3. Comparison of WT and 3′F mice at extremely advanced age.**
**(A)** Long-term survival curves of WT versus 3′F mice. Mantel–Cox log-rank test. n = 20 of each sex for each genotype. **(B)** Representative image of spleens from mice of indicated genotype. **(C)** Quantitation of splenic masses. Spleens used in photo indicated by red data points. WT: n = 8; 3′F: n = 8. **(D)** Representative H&E–stained histological sections of liver and kidneys harvested from mice of indicated genotype. **(E)** Quantitation of histopathological index scores from indicated organs from mice of indicated genotype. Error bars = SD. **(F)** ELISA quantification of serum anti-dsDNA immunoglobulin. WT: n = 8; 3′F: n = 8. Error bars = SD. **(G)** Representative histograms showing splenic T-cell staining for CD69, CD62L, and CD44. **(H)** Representative flow cytometry data illustrating the comparative size of the splenic GR-1+CD11b+ myeloid compartment in animals of indicated genotype. **(I)** Quantitative comparison of myeloid compartment size. The dot plots shown are represented by red data points. WT: n = 7; 3′F: n = 7. All analyses performed on biological samples harvested from mice aged 816 ± 27 d. Statistical comparisons made using unpaired two-tailed *t* test unless otherwise noted. Error bars = SEM unless otherwise noted.

## *Mir-146a*–deficient, but not *mir-146a*$^{3'F/3'F}$ mice, show hyper-responsiveness to LPS challenge

Our findings above did not rule out the possibility that some difference in function might be observed by comparing the *mir-146a* and *mir-146a*$^{3'F}$ alleles in the context of challenge of the innate immune machinery, either *in vivo* or *in vitro*. We thus assessed this possibility, again closely following the previously established experimental workflow. As previously described, following sublethal challenge via injection of LPS (1 mg/kg) *mir-146a*–deficient mice exhibited elevated levels of serum TNFα, IL-1β, and IL-6 as compared with wild-type controls (Fig 5A). Analogous results were observed after challenge with a lethal (35 mg/kg) dose of LPS—*mir-146a*–deficient mice were characterized by elevated levels of serum TNFα, IL-1β, and IL-6 and succumbed to the lethal LPS dose at a significantly greater rate than wild-type controls (Fig 5B and C). However, although modest trends in cytokine production were observed in both LPS challenges, once again, responses in mice homozygous for the *mir-146a*$^{3'F}$ allele were statistically indistinguishable from their respective controls at the level of cytokine production and viability (Fig 5A–C). Furthermore, repetition of these experiments with *mir-146a*$^{+/−}$ and *mir-146a*$^{3'F/−}$ mice revealed no significant differences in circulating cytokine levels or LPS sensitivity (Fig S5). Altogether, these results indicate that the *mir-146a*$^{3'F}$ allele is essentially functionally equivalent to the wild-type *mir-146a* allele in these contexts.

To compare the function of the mutant and wild-type alleles in a cellular context, we challenged BMDMs from each line with LPS ex vivo. Again, consistent with previously published results, BMDMs from *miR-146a*–deficient mice were characterized by comparatively increased production of IL-6 at 8 and 16 h following the LPS challenge (Fig 5D). This response coincided with increased relative expression of IRAK1 protein beginning at 16 h post-challenge, consistent with the *Irak1* mRNA's established role as a target of miR-146a (Fig 5E and F). Interestingly, we did not observe dysregulation of TRAF6 in any of our in vitro experiments (data not shown). Although we noticed a modest trend for increased production of IL-6 by LPS-challenged *mir-146a*$^{3'F/3'F}$ BMDMs, this did not reach statistical significance (Fig 5D), and even this trend was significantly reduced when levels of IRAK1 protein were assessed (Fig 5E and F). Taken together, these data suggest that the *mir-146a*$^{3'F}$ allele may be modestly hypomorphic compared with the wild-type allele but not to a degree that can be readily demonstrated.

### Functional definition of miR-146a target spectrum

Having phenotypically assessed the function of the *mir-146a* allele within these contexts, we next wished to survey the transcriptome and proteome of *mir-146a*–deficient and *mir-146a*$^{3'F/3'F}$ BMDMs relative to wild-type controls. Three biological replicates of BMDMs derived from each of the three genotypes were stimulated with

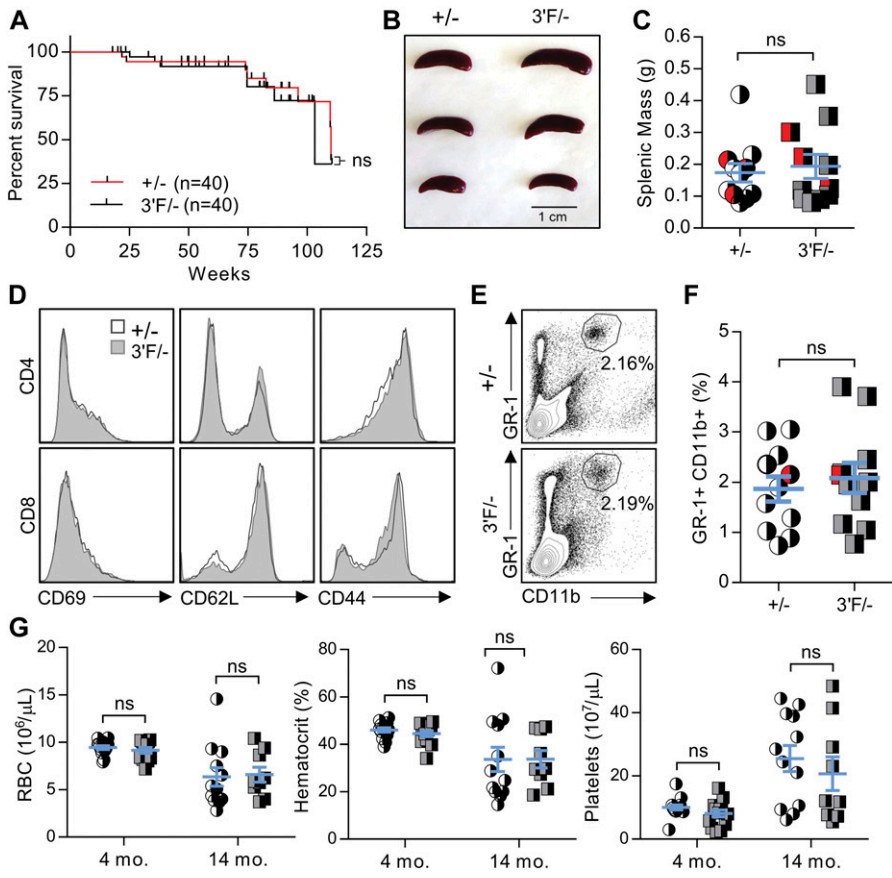

**Figure 4.** *Mir-146a$^{3'F/-}$* hemizygous mice do not display observable pathology.
**(A)** Long-term survival curves of +/− versus 3'F/− mice. Mantel–Cox log-rank test. n = 20 of each sex for each genotype. **(B)** Representative image of spleens from mice of indicated genotypes. **(C)** Quantitation of splenic masses. Spleens used in photo indicated by red data points. +/−: n = 11; 3'F/−: n = 11. **(D)** Histograms of splenic T-cell staining for CD69, CD62L, and CD44. **(E)** Representative flow cytometry data comparing percentage of the splenic GR-1$^+$CD11b$^+$ myeloid cells in animals of indicated genotype. **(F)** Quantitative comparison of myeloid compartment size. The dot plots shown are represented by red data points. +/−: n = 11; 3'F/−: n = 11. **(G)** Complete blood count parameters. +/− 4 mo: n = 14; 3'F/− 4 mo: n = 12; +/− 14 mo: n = 12; 3'F/− 14 mo: n = 9. All analyses performed on biological samples harvested from mice aged 217 ± 2 d unless otherwise indicated. Statistical comparisons made using unpaired two-tailed *t* test unless otherwise noted. Error bars = SEM.

1 ng/ml LPS for 16 h and harvested. Individual replicates were split into two fractions—one analyzed via next-generation sequencing and the other by label-free mass spectrometry proteome profiling.

Within mammalian systems, increased or decreased expression of a miRNA often correlates with a global shift in the relative expression of its predicted mRNA targets (Grimson et al, 2007). However, this characteristic shift was absent, not only in the *mir-146a$^{3'F/3'F}$* BMDMs, but even, surprisingly, in the *mir-146a*–deficient BMDMs when compared with wild-type controls. While differences of modest statistical significance were observed with lower target context scores (Grimson et al, 2007), the direction of these changes was opposite to what would be consistent with miRNA targeting, and no shift was observed for conserved miRNA targets (Fig 6A and Table S2). This apparent lack of response at the transcript level has been previously reported for miR-146a (Boldin et al, 2011). Because miRNAs in mammals can mediate translational blockade without any appreciable effect on mRNA expression, we also looked for this target shift at the proteomic level using mass spectrometry–based profiling. No discernible shift in predicted miR-146a targets was seen at the protein level in either *mir-146a$^{3'F/3'F}$* or *mir-146a*–deficient BMDMs either (Fig 6B and Table S3). Indeed, only one protein—IRAK1—that was predicted by TargetScan 7.1 (Agarwal et al, 2015) and/or present within miRDB (Wong & Wang, 2014) was found to be up-regulated relative to wild-type BMDMs in the *mir-146a*–deficient cells. Of note, IRAK1's expression was not significantly

different between wild-type and *mir-146a$^{3'F/3'F}$* BMDMs (Fig 5E and F and Table S3).

Despite the fact that predicted miR-146a targets did not broadly increase in relative expression in *mir-146a*–deficient BMDMs, we did identify several proteins that did, and also many whose expression decreased relative to wild-type controls (Table S3). We manually examined those proteins whose change in expression following LPS stimulation in the *mir-146a*–deficient BMDMs differed from those in wild-type by a Z score greater than three, and we did not find any statistically significant enrichment for MREs complementary to miR-146a. This was true whether we looked for a perfect predicted hexamer seed match or allowed for the possibility of G:U wobble pairing (data not shown). However, given that a small number of mRNA transcripts falling into the differentially up- and down-regulated groups did contain MREs that would be predicted to bind miR-146a (Fig S6), we subcloned the respective 3' UTRs or segments of the coding sequence (as appropriate) of these genes of interest into the *pRL-TK-CX6X* reporter to test their responsiveness to miR-146a and miR-146a$^{3'F}$ mimics. As a control, we also subcloned the 3' UTRs of strongly up- and down-regulated genes which lacked any recognized miR-146a target site. None of the predicted targets or negative controls responded to either mimic (Fig 7A, multiple *t* tests, FDR = 1%). Similar results were obtained when we assessed 3' UTRs of mRNAs containing miR-146a–complementary MREs and encoding proteins that differed in expression (Z score > 3) between wild-type and *mir-146a$^{3'F/3'F}$* BMDMs (Fig 7B). These

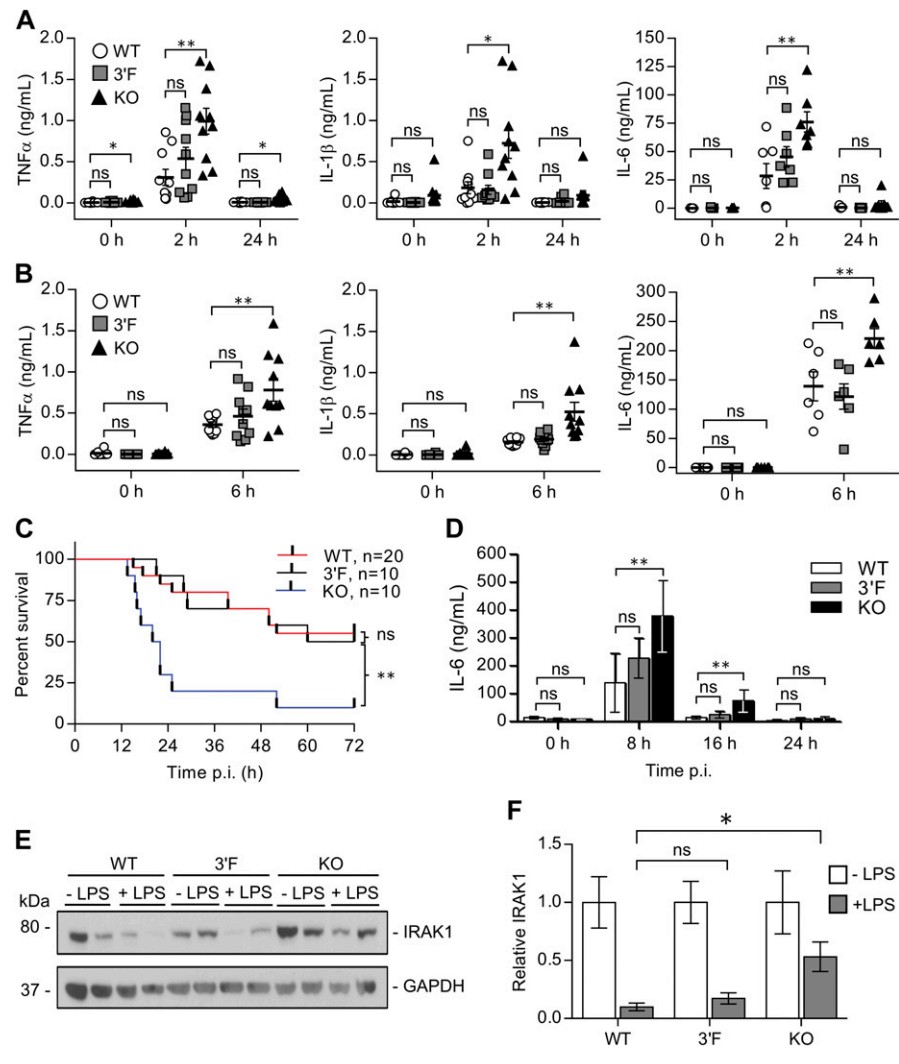

**Figure 5. The 3′F allele does not recapitulate the enhanced *in vivo* and *in vitro* response to LPS associated with *mir-146a* deficiency.**
**(A)** Quantitation of indicated serum cytokines from mice at indicated time points following sublethal (1 mg/kg) LPS challenge. **(B)** Quantitation of indicated serum cytokines from mice at indicated time points following lethal (35 mg/kg) LPS challenge. **(C)** Kaplan–Meier plot of mice survival following lethal (35 mg/kg) injection of LPS. Mantel–Cox log-rank test. Data in **(A)**, **(B)**, and **(C)** are pooled from three individual experiments. **(D)** ELISA quantitation of interval-based IL-6 expression from differentiated BMDMs cultured from each of the indicated genotypes and stimulated with 1 ng/ml LPS. n = 6 (WT), 6 (3′F), and 7 (KO). Error bars = SD. **(E)** Immunoblot of IRAK1 expression in BMDMs cultured from each of the indicated genotypes either unstimulated (−) or cultured for 16 h with 1 ng/ml LPS (+). Samples are run in biological duplicate. **(F)** Densitometric quantitation of IRAK1 blots, of which (E) is representative. For each biological sample, expression of IRAK1 was first normalized to its associated loading control. The resulting ratios were then graphed as a function of the unstimulated control's average expression for each genotype. Statistical comparisons were done using unpaired, two-tailed *t* test unless otherwise indicated. *P < 0.05; **P < 0.01 in all subpanels. Error bars = SEM unless otherwise indicated.

findings strongly argue against the possibility that differential protein expression of putative novel miR-146a and miR-146a³′F targets is a direct effect of differential targeting by wild-type and mutant miRNA.

## Discussion

We have characterized mutant mice in which the endogenous 3′ pairing of miR-146a, an established regulator of NF-κB and innate immune signaling, is disrupted. By carefully revisiting the battery of experiments initially used to characterize *mir-146a*–deficient mice, we demonstrate here that the *mir-146a³′F* allele recapitulates the function of wild-type miR-146a in each of the previously established phenotypes. Some nonsignificant trends towards the *mir-146a*–deficient phenotype were noted in a comparison of *mir-146a⁺/⁻* and *mir-146a³′F/⁻* mice and cells, and we were able to document a modest difference in dose–response to reporters under the control of the *Irak1* and *Traf6* 3′ UTRs. It is, thus, formally possible that the disruption of miR-146a's endogenous 3′ pairing results in an allele

that is very mildly hypomorphic, and that statistical significance would ultimately have been reached with greater numbers of mice. However, given that *mir-146a³′F/⁻* mice thus far (100 wk at the time of writing) display indistinguishable health and longevity to *mir-146a⁺/⁻* littermates, any decreased function appears to be quite modest.

One might be tempted to consider that the *mir-146a³′F* allele does, in fact, have impaired functionality, but that this is offset in mice of the relevant genotypes by increased relative expression/compensatory function of *mir-146b*, the sequence of which is highly similar to *mir-146a*. However, given that *mir-146b* wild-type function is genetically preserved in *mir-146a*–deficient mice, the phenotype associated with *mir-146a* deficiency is a product of the former's wild-type function and the latter's deficiency, respectively, and any genetic rescue of this phenotype can be most reasonably attributed to restoration of absent *mir-146a* function. Since the *mir-146a³′F* allele rescues *mir-146a* deficiency to a level statistically indistinguishable from the wild-type allele (even within the hemizygotic state), we favor the conclusion that this allele, from a genetic standpoint and in the context examined, is essentially and functionally

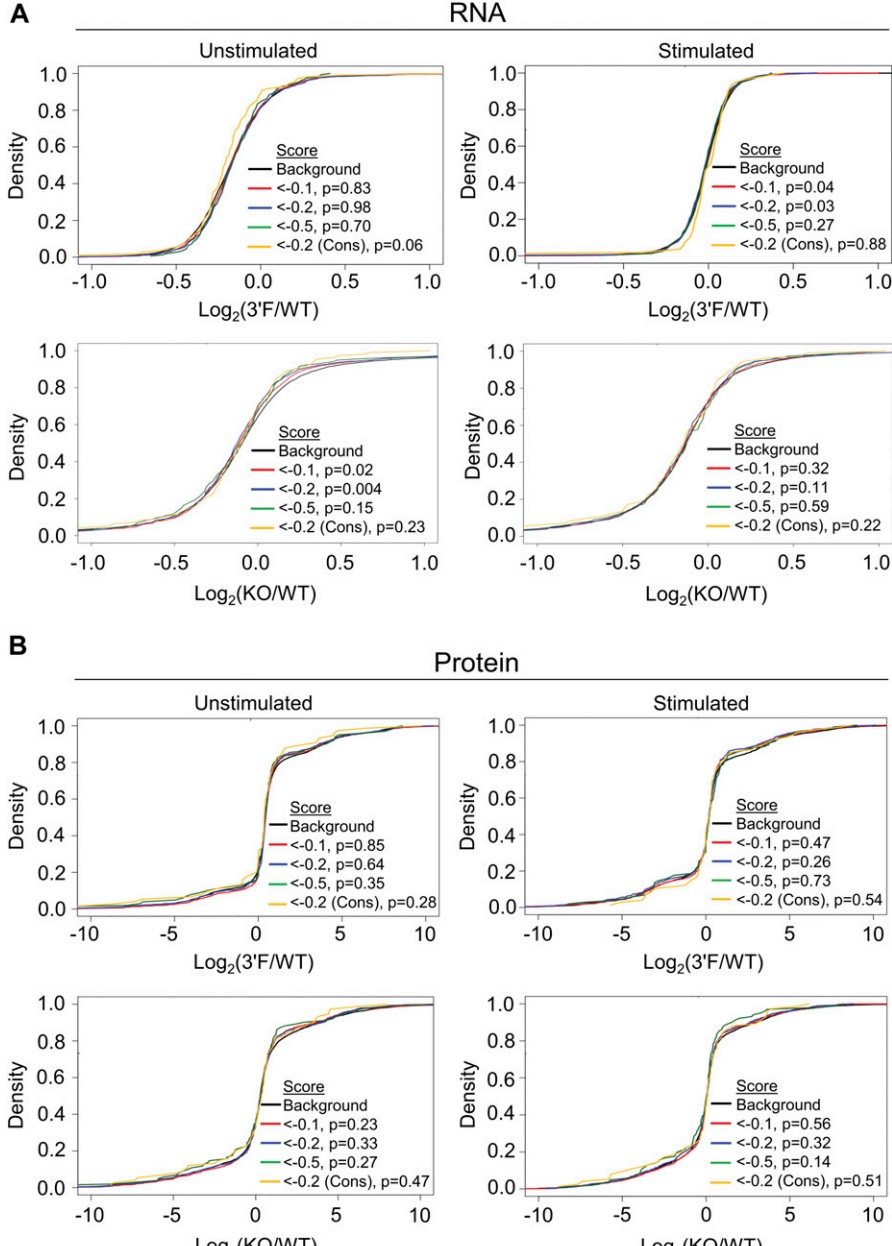

**Figure 6. Neither 3′F nor KO BMDMs display classical miRNA target perturbation at the transcriptional or protein level.**
**(A)** Cumulative distribution plots comparing the $\log_2$ fold-change of predicted mRNA targets of miRNA with that of background set (mRNAs lacking predicted miRNA target sites in their 3′ UTR). **(B)** Cumulative distribution plots comparing the $\log_2$ fold-change of predicted protein targets of miRNA with that of background set (mRNAs lacking predicted miRNA target sites in their 3′ UTR). Wilcoxon rank sum test.

equivalent to the wild-type *mir-146a* allele. Our small RNA sequencing data support this conclusion. Although miR-146b does appear to be elevated in cells derived from *mir-146a*–deficient mice, this elevation is not enough to maintain normal response to LPS stimulation in these cells, and no such elevation is observed in cells derived from mice homozygous for the *mir-146a$^{3'F}$* allele. If one were to speculate that this elevation of miR-146b in *mir-146a*–deficient mice was a product of homeostatic mechanisms pressuring compensatory function, one could then reasonably conclude that no such homeostatic pressure was present in the context of *mir-146a$^{3'F}$* allelic function.

Generally, our findings stand at apparent odds with both our initial *in vitro* data and with the large body of recent work demonstrating both that many miRNAs engage their targets via 3′ pairing and that disruption of this pairing can negatively impact the regulatory relationship between a given miRNA and its mRNA target (Chi et al, 2009; Hafner et al, 2010; Loeb et al, 2012; Helwak et al, 2013; Grosswendt et al, 2014; Moore et al, 2015; Broughton et al, 2016). Regarding the former, we feel our work suggests that caution is warranted in the evaluation of *in vitro* models and, more specifically, that differences observable in the artificial context of reporter assays may not, in fact, accurately reflect or recapitulate *in vivo* function. Concerning the many studies where contribution of the miRNA 3′ region has been described, we do not feel that any of the experiments in our study directly challenge or invalidate the observations described in these previous works. Indeed, a consistent

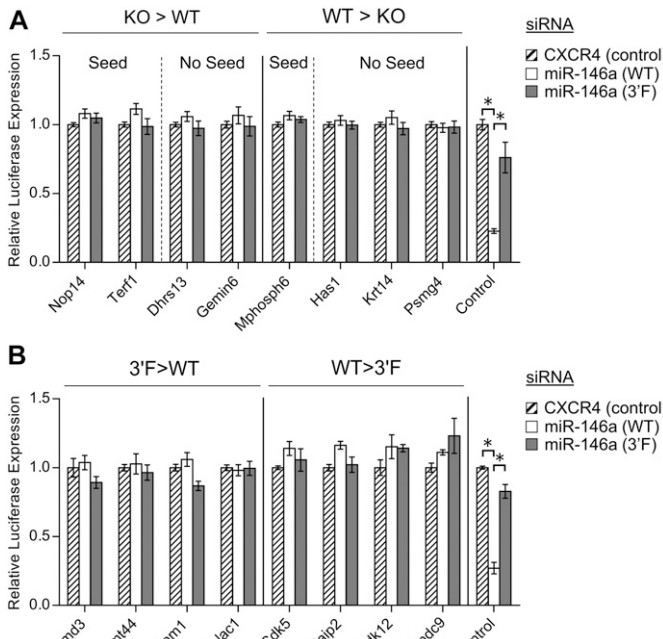

**Figure 7. Targets differentially expressed during BMDM stimulation as a function of genotype are not direct miR-146a or miR-146a$^{3'F}$ targets.**
**(A)** Dual luciferase assays of putative miR-146a targets characterized by differential protein expression following LPS stimulation in a comparison of KO and wild-type BMDMs. Left: proteins with reduced expression in response to LPS in the wild-type BMDMs as compared with the KO BMDMs; right: regions derived from proteins with reduced expression in response to LPS in the KO BMDMs as compared with the wild-type BMDMs. **(B)** Dual luciferase assays of putative miR-146a targets characterized by differential protein expression following LPS stimulation in a comparison of 3'F and wild-type BMDMs. Left: proteins with reduced expression in response to LPS in the wild-type BMDMs as compared with the 3'F BMDMs; right: regions derived from proteins with reduced expression in response to LPS in the 3'F BMDMs as compared with the wild-type BMDMs. In both of these experiments, the control reporters to the far right are the same as those described in Fig S1—synthetic reporters with six sequential MREs corresponding to the wild-type miR-146a sequences. Statistical comparisons were done using multiple *t* test. For initial selection, false discovery rate was set at 1%. * denotes statistical significance. No relationships are statistically significantly different unless noted. Error bars = SEM.

theme of several of these studies is that miRNAs employ a diversity of target-pairing strategies. Thus, it remains entirely plausible that one particular miRNA might be entirely independent of 3' binding, although any number of others may use such binding as a supplementary or even necessary component of their function. This notion may also be applied to interactions within a given miRNA's complement of physiological targets. It is also important to note that although characterization of a murine allele allows for a comparatively robust assessment of multiple distinct phenotypes, this approach precludes the more systematic analyses that are possible in other model organisms or cell-based systems. We, therefore, consider it unwise at this juncture to make broad conclusions about the importance of miRNA 3' pairing in mammalian systems until additional work is made available.

A final and somewhat unexpected revelation from our study is our inability to detect a global shift in the relative expression of predicted miR-146a targets in either *mir-146a*–deficient or

*mir-146a*$^{3'F/-}$ bone marrow–derived macrophages, whether at the level of the transcriptome or the proteome. Although similar conclusions have been drawn using different methods in an earlier analysis of the transcriptome of *mir-146a*–deficient mice (Boldin et al, 2011), we were surprised that *Irak1* was the only predicted miR-146a target whose changes in protein expression were consistent with *mir-146a* loss-of-function and rescue by the *mir-146a*$^{3'F}$ allele. Interestingly, although heightened expression of TRAF6 protein was previously reported for *mir-146a*–null mice (Boldin et al, 2011), we did not observe this in any of our in vitro experiments (data not shown). However, this disconnect is not entirely without precedent in the literature. *Traf6* mRNA levels have been reported to be unperturbed by miR-146a in LPS-stimulated B cells (Jiang et al, 2018) and in the PBMCs of patients with Sjögren's syndrome (in which miR-146a levels are elevated), where *Traf6* mRNA levels are, paradoxically, abnormally high (Zilahi et al, 2012). Although admittedly the abundance of a particular transcript is certainly not an indication of its active translation, these results taken together may suggest miR-146a's established role in regulating *Traf6*—and perhaps other genes—is more context dependent than generally assumed. Beyond this, although the observation that the complement of predicted miR-146a target mRNAs were not characterized by any significant aggregate shift in abundance in the comparison of *mir-146a*–deficient and wild-type BMDMs would seem incongruous with a "micro-manager"–like regulatory role for miR-146a in this context, this regulatory role cannot be formally excluded with the data we present. Nonetheless, the aggregate of our data, when considered alongside the extant body of work on this general topic, support a nuanced view of miRNA target spectrum and mechanism of action that may vary upon context, whether this context is species, the identity of the miRNA and its target, or cell state.

# Materials and Methods

### Generation of *mir-146a* KO and 3'F mice

A ~7.1-kb region of the *Mus musculus* genome from chromosome 11 was amplified in two arms. Arm 1 (5.1 kb) was amplified from position 43,373,405 to 43,378,474 using primers *146_arm1_fw* (5'-atcgta*aagctt*CCCAGGTACTGGGAAGAACA-3') and *146_arm1_rev* (5'-CACCTCAGCAGAC CATGCTA-3'). Arm 2 (2.1 kb) was amplified from position 43,371,337–43,373,508 using primers *146a_arm2_fw* (5'-atcgta*ggatcc*GAGAGACACAGGATTGCCAAGCAGTGATTTC-3') and *146a_arm2_rev* (5'-atcgta*gcggccgc*CACTGGCTAAGGGTCGGATA-3'). Arm 1, containing the *mir-146a* locus, was cloned into the *HindIII* site of the *pIDT Smart* ampicillin vector (IDT), and the WT *mir-146a* sequence was removed by cutting out a 536-nt region flanked by a *BspHI* recognition site at the 5' end and a *BclI* recognition site at the 3' end, then substituting in commercially synthesized sequences (IDT) either lacking the *mir-146a* pre-miRNA sequence entirely or comprising the 3'F mutation.

Arm 2 (5' *BamHI* and 3' *NotI*) and the edited Arm 1 (*HindIII*) were then subcloned into the *pBluescript II KS(+)* (Addgene) vector on either side of a *loxP*-flanked region containing the neomycin resistance cassette. The resulting *mir-146a*$^{3'F}$ and *mir-146a*$^-$ targeting

constructs were electroporated into V6.5 murine embryonic stem cells, and the colonies were grown and picked under standard G418/ganciclovir double selection. Colonies derived from correctly recombined cells were identified via Southern analysis as described (Neilson et al, 2004), double digesting the genomic DNA with *BamHI* and *EcoRV* to detect insertion of an *EcoRV* site derived from the polylinker of the targeting vector. The Southern probes, amplified from genomic C57/BL6 genomic DNA corresponded to chr11:43,377,851–43,378,550 and chr11:43,370,951–43,371,356 (both GRCm38/mm10).

The G418 selection cassette was removed via transient transfection of *Cre* recombinase, and embryonic stem cells in which the cassette had been excised were identified via replicate plating in the presence and absence of G418. Correctly targeted embryonic stem cells were injected into C57BL6/J blastocysts to obtain chimeric mice and germline transmission. Heterozygous animals were intercrossed to obtain both *mir-146a*$^{-/-}$ and *mir-146a*$^{3'F/3'F}$ homozygotes.

### Animal studies

Transgenic founder animals from each line were individually backcrossed three times to the C57BL6/J line (Jackson) and breeding colonies were separately maintained for each line, using heterozygotes for each mutant allele to produce paired homozygous mutant and wild-type control animals for each line. Age- and sex-matched mice from each of the two lines were used for all experiments (except where the use of sex-matched littermates is explicitly noted) such that homozygous mutants from each line were compared with an equivalent number of wild-type mice equally derived from the same two lines. No differences were noted in the phenotypic comparison of wild-type mice derived from breeding of *mir-146a*$^{+/-}$ heterozygotes and wild-type mice derived from breeding of *mir-146a*$^{3'F/-}$ heterozygotes in any experiments. All mouse procedures were approved by the Institutional Animal Care and Use Committees of the Baylor College of Medicine.

### Direct quantitation of WT and 3′F miR-146a

Liver, thymus, kidney, lymph nodes, spleen, and whole bone marrow were excised from age- and sex-matched WT, *mir-146a*–deficient, and *mir-146a*$^{3'F/3'F}$ mice and physically disaggregated. Splenocytes were separated into B cells and CD4$^+$ and CD8$^+$ T cells via flow sorting using a FACSAria II (BD). Bone marrow cells were cultured for 1 wk in either IL-4 (10 ng/ml) + GM-CSF (30 ng/ml) or M-CSF (10 ng/ml) (all R&D Systems) for differentiation to either BMDCs or BMDMs, respectively. Differentiation was confirmed via flow cytometry using an LSRFortessa (BD). BMDMs and BMDCs were stimulated with 1 ng/ml LPS for 16 h. RNA was isolated from all samples using TRIzol LS reagent (Life Technologies) as per the manufacturer's recommendations and quantified by spectrophotometry. Commercially obtained WT and 3′F miR-146a mimic direct quantification standards were serially diluted in 1 µg/ml total RNA derived from *Drosophila* S2 cells. Reverse transcription was performed with the miScript II RT kit (QIAGEN) and qPCR was performed using the miScript SYBR Green PCR kit (QIAGEN), both according to the manufacturer's instructions. Absolute miRNA quantification

was calculated based on total input RNA for each biological sample. Primers specific for either the WT or the 3′F allele were used on all samples to ensure accurate genotyping and to rule out cross-priming.

### Luciferase reporter constructs and luciferase assay

The 3′ UTRs of indicated genes were amplified from mouse genomic DNA via PCR and subcloned into an edited version of the *pRL-TK-CX6X Renilla* luciferase reporter plasmid (Addgene) via *XhoI* and *ApaI* or *XhoI* and *NotI* sites added to the 5′ and 3′ ends of the PCR amplicons. Following restriction digestion, amplicons were ligated with the vector using the NEB Quick Ligation kit (NEB). Predicted miR-146a recognition sites within the UTRs were mutated as indicated in the text using the QuikChange II site-directed mutagenesis kit (Thermo Fisher Scientific). A more detailed catalog of oligonucleotide sequences is provided in Table S4.

HeLa cells were seeded in 24-well plates at a seeding density of 40,000 cells/well in DMEM supplemented with L-glutamine, penicillin, streptomycin, sodium pyruvate, Hepes (pH 7.4), nonessential amino acids, and β-mercaptoethanol (DMEM complete) and allowed to grow overnight. The following day, the cells were transfected with the pGL3 *Firefly* luciferase control plasmid (Promega), one of the experimental *Renilla* luciferase reporter plasmids, and varying concentrations of either the WT or 3′F miRNA mimic duplexes (Sigma-Aldrich), along with a balanced amount of irrelevant anti-CXCR4 siRNA duplex (Sigma-Aldrich) to ensure that the final siRNA concentration in each well was always equal to 31.6 nM. Transfections were performed using the Lipofectamine 3000 transfection reagent system (Thermo Fisher Scientific) as per the manufacturer's protocol. Luciferase assays were performed using the Dual-Luciferase Reporter Assay System (Promega) as per the manufacturer's protocol on a Tecan M200 multimode reader using Tecan Magellan software (Tecan).

### Splenic immunophenotyping

Spleens were excised, massed, photographed, and disaggregated. Red blood cells were lysed with ACK buffer, and the splenocyte suspensions were quantified before staining for the indicated markers. One million splenocytes were stained using a 1:100 dilution of fluorophore-tagged antibodies in 100 µl PBS + 1% FBS + 1 mM EDTA for 15 min at 4°C, after which the cells were washed three times. Cell viability was ensured by staining with NucBlue Live ReadyProbes Reagent (Thermo Fisher Scientific). Cells heat-killed at 65°C for 15 min were used as a control. Data were collected using a BD LSRFortessa flow cytometer and analyzed using Kaluza Analysis software V1.5 (Beckman Coulter). A detailed catalog of antibodies utilized is provided in Table S4.

### Histological analysis

Livers and kidneys were excised to 70% ethanol and submitted to the Baylor College of Medicine Center for Comparative Medicine, where they were processed for hematoxylin and eosin staining via standard methods. Histological scoring was performed by a staff licensed veterinary pathologist blinded to the origin of the histological samples.

## LPS challenges

For sublethal LPS challenges, mice were bled retro-orbitally and then stimulated with LPS at a dose of 1 mg/kg body mass. The mice were bled again at 2 h and 24 h post-injection. For lethal LPS challenges, the mice were retro-orbitally bled and then stimulated with LPS at a dose of 35 mg/kg body mass. The mice were bled again at 6 h post-injection. Serum was collected from the blood via differential centrifugation at 9,800 $g$ for 7 min at 4°C. IL-6 was quantified via commercial ELISA kit (R&D Systems) following the manufacturer's instructions. Other cytokines were quantified via Luminex assay (Thermo Fisher Scientific) by the Baylor College of Medicine Antibody-Based Proteomics core. For lethal LPS challenges, the mice were monitored hourly for morbidity beginning at 10 h post-injection.

## CBC analysis

Blood was collected via retro-orbital bleeding into EDTA-coated collection vials (BD Pharmingen) and submitted to the Baylor College of Medicine Center for Comparative Medicine.

## Isolation and culture of BMDMs

Bone marrow from 6- to 7-mo-old mice was flushed from femurs and tibias and plated at 85,000 cells/cm$^2$ on non–tissue culture–treated petri dishes. The cells were cultured for 8–9 d in DMEM complete medium supplemented with M-CSF (10 ng/ml), with a medium change at day 3. Successful differentiation was verified by surface staining of CD11b, CD11c, F4/80, CD86, and GR-1, followed by flow cytometry analysis as described above.

## In vitro BMDM LPS challenges

BMDMs were stimulated with 1 ng/ml LPS for 24 h. To allow monitoring of cytokine production specifically within various windows following stimulation, cell culture medium was collected and replaced with medium supplemented with 1 ng/ml LPS every 8 h following stimulation. The culture medium was snap-frozen in liquid nitrogen and stored at −80°C. Analysis of IL-6 content in the medium was carried out via commercial ELISA kit (R&D Systems) according to the manufacturer's instructions.

## Immunoblots

BMDMs were lysed in RIPA buffer (10 mM Tris [pH 7.4], 150 mM NaCl, 1% Triton X-100, 0.1% SDS, 0.5% Na-Desoxycholate, 1 mM EDTA, and 1 mM DTT) containing protease and phosphatase inhibitors, snap-frozen, and stored at −80°C. Lysate protein concentration was quantified by the BCA protein assay. 40 $\mu$g of sample was diluted in 1× RIPA buffer, denatured in Laemmli buffer at 90°C for 10 min, and loaded into a 4–12% polyacrylamide gel. The gels were transferred to a PVDF membrane at 30 V for 1 h, blocked in 10% non-fat milk in TBST, and then stained overnight at 4°C with primary anti-IRAK1 (CST) and anti-GAPDH antibodies (EMD Millipore). The next day, the blots were washed three

times with TBST, then incubated in HRP-conjugated secondary antibody for 30 min at room temperature. Blots were developed with ECL 2 solution (Thermo Fisher Scientific) and exposed to film. Again a detailed catalog of antibodies utilized is provided in Table S4.

## RNA Seq

Total RNA was extracted using Trizol (Thermo Fisher Scientific) according to the commercial protocol. 100 ng of total RNA was used for rRNA depletion using Ribo-Zero Gold rRNA removal Kit (Illumina) and following a modified protocol for low inputs. Half of the rRNA-depleted RNA was used for library preparation using NEBNext Ultra II Directional RNA Library Prep Kit (New England Biolabs), followed by sequencing on a NextSeq500 instrument (Illumina). The raw reads were processed to remove the adapter sequences using cutadapt (v1.8.3) with the following arguments: – a GATCGGAAGAGCACACGTCTGAACTCCAGT, −m 20. Following adapter trimming, the reads originating from rRNA were removed by aligning the remaining reads to the rDNA using HISAT2 (v2.1.0) with the following arguments: −−no-spliced-alignment, −−un. Reads that aligned to the rDNA were discarded. Remaining reads were aligned to the mouse genome (mm10) using HISAT2 (v2.1.0) with the following arguments: −−min-intronlen 70, −−rna-strandness R, −−known-splicesite-infile. The raw read counts were obtained for each gene (Gencode's comprehensive gene annotation release M15) using featureCounts (v1.5.3) with the following arguments: −s 2, −Q 50. For the miRNA targeting analysis, log$_2$ fold-change of predicted targets of miR-146 (TargetScanMouse v7.1) was compared with the log$_2$ fold-change of background set (mRNAs that do not contain miRNA target sites in their 3′ UTR). The significance of the difference was evaluated using Wilcoxon rank sum test. Small RNA sequencing was performed essentially as described (Wissink et al, 2016). Raw RNAseq data may be obtained via NCBI GEO accession number GSE125474.

## Mass spectrometry

Mass spectrometric analysis was carried out on an Orbitrap Fusion LumosETD (Thermo Fisher Scientific) by the Mass Spectrometry Proteomics core at Baylor College of Medicine as described before (Saltzman et al, 2018). Briefly, BMDM lysates from 8- to 12-wk-old mice were denatured in ABC solution (50 mM ammonium bicarbonate + 1 mM CaCl$_2$), with subsequent snap-freeze/42°C thaw cycling and boil denaturing at 95°C. The proteins were digested with a 1:20 solution of 1 $\mu$g/$\mu$l trypsin:protein overnight at 37°C with shaking, and then with a 1:100 solution of 1 $\mu$g/$\mu$l trypsin:protein for 4 h. The peptides were extracted and measured using the Pierce Quantitative Colorimetric Peptide Assay (Cat. No. 23275; Thermo Fisher Scientific). Next, 50 $\mu$g of vacuum-dried peptides were re-dissolved in pH10 ABC buffer (10 mM ammonium bicarbonate, pH 10, adjusted with NH$_4$OH) and subjected to off-line microscaled reverse-phase separation on a micropipette tip layered with 6 mg of C18 matrix (Reprosil-Pur Basic C18, 3 $\mu$m, Dr. Maisch GmbH) on top of a C18 disk plug (EmporeTM C18, 3M). A total of 15 (2–30% ACN, 2% steps) fractions were obtained and combined into five pools for mass spectrometry sequencing (15F5R protocol with 02+12+22, 04+14+24, 06+16+26, 08+18+28, and 10+20+30% combinations). For each

run on the Lumos instrument, ~1 µg of peptide was loaded onto a 2-cm 100 µm ID pre-column and resolved on a 6-cm 150 µm ID column, both packed with sub-2 µm C18 beads (Reprosil-Pur Basic C18, Cat. No. r119.b9.0003, Dr. Maisch GmbH). The gradient mobile phase was mixed from water (solution A) and 90% acetonitrile (solution B), both with 0.1% formic acid. A constant flow rate was maintained with 75-min linear gradient elutions. The Proteome Discoverer software suite (PD version 2.0.0.802; Thermo Fisher Scientific) was used to search the raw files with the Mascot search engine (v2.5.1, Matrix Science), validate peptides with Percolator (v2.05), and provide MS1 quantification through Area Detector Module. MS1 precursors in a 350–10,000 mass range were matched against the tryptic RefProtDB database digest with Mascot permitting up to of two missed cleavage sites (without cleavage before P), a precursor mass tolerance of 20 ppm, and a fragment mass tolerance of 0.5 D. The following dynamic modifications were allowed: acetyl (protein N-term), oxidation (M), carbamidomethyl (C), DeStreak (C), and Deamidated (NQ). For the Percolator module, the target strict and relaxed FDRs for PSMs were set at 0.01 and 0.05 (1 and 5%), respectively. gpGrouper was used to assemble peptide identifications into gene products, and the final proteome data matrix were filtered to proteins with at least two identifications within a treatment/genotype triplicate. Protein groups were quantified by iBAQ values and median-normalized. Raw mass spectrometry data may be obtained from the PRIDE partner repository under accession number PXD011413.

**Putative novel miR-146a target identification**

The difference in $\log_2$ fold-change of proteins upon LPS stimulation in the mass spectrometry analysis was calculated for WT and KO (and separately, WT and 3′F) BMDMs. The respective lists were sorted on this difference, and candidate targets were drawn from two pools of gene products with a Z score of >3 in the positive or negative direction from the median. Candidate novel targets were defined as those with a putative seed in the CDS or 3′ UTR of the corresponding mRNA and characterized by a significant change (q < 0.05) in one or both of the genotypes compared. These candidates (along with negative controls characterized by differential expression but no predicted seed region) were amplified from murine genomic DNA via PCR and inserted into the *pRL-TK-CX6X* backbone, as described above.

**Statistics**

For comparisons between two groups, statistical significance was assessed using a two-tailed *t* test both with and without assumption of equal variance. Neither assumption impacted the level (or lack thereof) of significance depicted in any representation. Data are presented as mean ± SEM in pooled experiments and mean ± SD in representative individual experiments. Comparisons of different miRNA mimics across multiple concentrations were analyzed by two-way ANOVA. Survival curves were evaluated using Mantel–Cox log-rank test. In all cases, * denotes a *P*-value of <0.05 and ** denotes a *P*-value of <0.01. All statistical analyses were performed using GraphPad Prism v6.0.

# Supplementary Information

# Acknowledgements

We wish to acknowledge Baylor College of Medicine's Advanced Technology Core Facilities, Center for Comparative Medicine, and the laboratories of William Lagor, William Decker, Eric Wagner, and Koen Venken for materials, use of equipment, and technical assistance. We thank Jennifer Grenier (RNA Sequencing Core, Cornell University) for RNA library preparation and sequencing. Research reported in this publication was supported by institutional funds from Baylor College of Medicine; the Nancy Chang Foundation; NIH CA190467 (to JR Neilson); T32AI053831 and T32DK060445 (to G Bertolet); P30AI036211, P30CA125123, and S10RR024574 (supporting the Cell Sorting Core); P50HD076210 (supporting the RNA sequencing Core); and P30CA125123 and CPRIT RP170005 (supporting the MS Proteomics Core). JR Neilson is the Athena Water Breast Cancer Research Scholar of the American Cancer Society (RSG-15-088-01RMC). The content is solely the responsibility of the authors and does not necessarily represent the official views of the National Institutes of Health.

## Author Contributions

G Bertolet: formal analysis, validation, investigation, methodology, and writing—original draft.
N Kongchan: resources, investigation, and writing—review and editing.
R Miller: data curation, investigation, and writing—review and editing.
RK Patel: data curation, investigation, and writing—review and editing.
A Jain: formal analysis, investigation, and methodology.
JM Choi: formal analysis, investigation, and methodology.
A Saltzman: formal analysis, investigation, and methodology.
A Christenson: formal analysis, investigation, and writing—review and editing.
SY Jung: conceptualization, formal analysis, supervision, investigation, and methodology.
A Malovannaya: conceptualization, formal analysis, supervision, investigation, methodology, and writing—review and editing.
A Grimson: conceptualization, formal analysis, investigation, methodology, and writing—review and editing.
JR Neilson: conceptualization, formal analysis, supervision, funding acquisition, validation, investigation, visualization, methodology, project administration, and writing—review and editing.

### Conflict of Interest Statement

The authors declare that they have no conflict of interest.

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
