## [Reviewer comments · Life Science Alliance]

Life Science Alliance

MiR-146a wild-type 3' sequence identity is dispensable for proper innate immune function in vivo

Grant Bertolet, Natee Kongchan, Rebekah Miller, Ravi Patel, Antrix Jain, Jong Choi, Alexander Saltzman, Amber Christenson, Sung Jung, Anna Malovannaya, Andrew Grimson, and Joel Neilson
DOI: <https://doi.org/10.26508/lsa.201800249>

Corresponding author(s): Joel Neilson, Baylor College of Medicine

Review Timeline:	Submission Date:	2018-11-19
	Editorial Decision:	2018-12-18
	Revision Received:	2019-01-15
	Editorial Decision:	2019-01-28
	Revision Received:	2019-01-31
	Accepted:	2019-02-01

Scientific Editor: Andrea Leibfried

Transaction Report:

December 18, 2018

Re: Life Science Alliance manuscript #LSA-2018-00249

Dr. Joel R. Neilson
Baylor College of Medicine, Houston, TX
One Baylor Plaza MS BCM335
Room S444
Houston, Texas 77030

Dear Dr. Neilson,

Thank you for submitting your manuscript entitled "MiR-146a wild-type 3' sequence identity is dispensable for proper innate immune function in vivo" to Life Science Alliance. The manuscript was assessed by expert reviewers, whose comments are appended to this letter.

As you will see, your work received split views. While reviewer #1 is supportive of further consideration of a revised version here, reviewer #2 and #3 have reservations about the value of the dataset provided. More specifically, reviewer #2 thinks that the miRNA chosen for your analysis would need to be clearly linked to a particular target in the context of your work to be able to arrive at strong conclusions. Reviewer #3 thinks that the differences in developing a phenotype in mice may be due to redundancy of miR-146a and 146b, the latter compensating in presence of the 3' mutated miR-146a but not in the full KO case. This reviewer also thinks that the proteome and transcriptome analysis should be performed in different cells than currently done.

Given this divergent input, I have discussed your work further with our editorial board, and we see value in your findings and decided to invite you to submit a revised version of your manuscript. We noted the prior work linking (partially) the KO phenotype to TRAF6 abundance (hence at least partially already addressing the major concern of reviewer #2). Providing a rescue experiment in the context of this work would in our view be great, but seems out of scope of a normal revision and is thus not expected. But the limitations noted by reviewer #2 and #3 should get discussed, and alternative explanations should get clearly mentioned as possible scenarios. Furthermore, the minor points of reviewer #2 should get addressed and, importantly, all the technical issues noted by reviewer #1 as well as the requests for clarification need to get addressed in the revised version. Please also explain the issue noted by reviewer #2, minor point 1.

Thank you for this interesting contribution to Life Science Alliance. We are looking forward to receiving your revised manuscript.

Sincerely,

- A letter addressing the reviewers' comments point by point.
- An editable version of the final text (.DOC or .DOCX) is needed for copyediting (no PDFs).
- High-resolution figure, supplementary figure and video files uploaded as individual files: See our detailed guidelines for preparing your production-ready images, <http://life-science-alliance.org/authorguide>
- Summary blurb (enter in submission system): A short text summarizing in a single sentence the study (max. 200 characters including spaces). This text is used in conjunction with the titles of papers, hence should be informative and complementary to the title and running title. It should describe the context and significance of the findings for a general readership; it should be written in the present tense and refer to the work in the third person. Author names should not be mentioned.

B. MANUSCRIPT ORGANIZATION AND FORMATTING:

Full guidelines are available on our Instructions for Authors page, <http://life-science-alliance.org/authorguide>

Reviewer #1 (Comments to the Authors (Required)):

In this manuscript, G. Bertolet and colleagues investigate the functional consequences of mutating the 3'-most sequence of miR-146a, without changing the seed nor the central region. Because conflicting data has been published regarding the contribution of non-seed regions, it is indeed important to probe the consequences of such a mutation (especially when the tested phenotypes include *in vivo*, integrated readouts, as is the case here). The authors find that the mutation triggers some consequences at the microscopic level (when the readout is a target RNA or protein abundance), but they did not find any convincing proof for macroscopic consequences. This is an interesting manuscript, addressing an important question in the field of miRNA-mediated gene regulation. The detailed phenotypic assessment (including many physiological assays, interrogating both homozygous and hemizygous mutants) is exemplary. Several problems (two major points, which seem to affect the authors' interpretations; but also many minor points, which make the manuscript obscure or inconsistent on a few important issues) tend to weaken its conclusions, so I recommend correcting all these points. Once this is done, the manuscript will be a strong candidate for publication in Life Science Alliance.

Major points:

1. Page 5, sentence "Transfection of this vector into HEK 293 cells [...]": to me, Fig. 1A does not support such a conclusion. It is not exactly clear to me what the authors mean with "similarly processed". Do they imply that the mutant miRNA is matured from Drosha and Dicer cleavages occurring at the same positions? If yes, that poorly-resolved Northern blot, showing smeary bands on two distinct membranes (none of them containing a size marker), does not provide enough information on miRNA length to conclude anything (not even mentioning the fact that miRNAs can have the same length but not the same 5' and 3' ends). Or do the authors mean that the mutant miRNA accumulates at the same level than the wt miRNA? If yes, these two membranes, hybridized with two different probes, are also useless for such a conclusion. It is actually possible to use Northern blot for such an assessment, but then each membrane has to be calibrated with synthetic RNA oligos in a few additional lanes, in order to convert Northern blot intensity signals into fmol of RNA (apparently Fig. 1G used calibrating synthetic oligos: these very same oligos could be used to make a Northern blot really quantitative for Fig. 1A).

2. Page 28: Fig. 1H compares WT to 3'F in the left panel, and WT to KO in the right panel. Why are the WT histograms so different between the two panels? This is not just a normalization issue, because even the ratios between samples are clearly different (e.g., compare thymus to spleen in both panels). It may be that the PCR-based quantification is not that precise (that would be perfectly understandable), or that there is a lot of biological variability between the batches of mice. But then, why are the s.e.m. so narrow? Whether there is a lot of technical noise, or a lot of biological variability, this should also reflect in the s.e.m. error bars... In the end, if WT replicates are so poorly reproducible, then the whole assay may not be able to detect efficiently any potential difference between WT and 3'F miRNAs.

Minor points:

1. Page 2, sentence "The first unbiased identification of bona fide miRNA/mRNA target pairs [...]": I would not agree that the results shown in Chi et al. (2009) "largely corroborated" computational predictions. The Chi et al. paper is certainly not very precise regarding the number and identity of CLIP-identified miRNA targets, which does not ease comparison to computational predictions - but at least their number of targets per miRNA, and the high percentage of seedless targets they found, seem to argue against such an optimistic affirmation. Please justify that sentence.
2. Page 3, sentence "This notion would partially explain the strict evolutionary conservation [...]". Another phenomenon seems to explain equally well the conservation of non-seed sequences: because miRNA*s seem to be under a selective pressure to repress seed-matched transcripts (see <https://www.ncbi.nlm.nih.gov/pubmed/21177881>). As the pre-miRNA secondary structure (hairpin-shaped) is also conserved, then the miRNA sequence facing the miRNA* seed has to be conserved - implying that the miRNA 3' end should be conserved even if it did not play any function in miRNA target recognition.
3. Page 4: for the sake of clarity, the sequence of wild-type and 3' F miR-146a should be given in a main figure (they are currently given in supplementary figure 2, which does not make it easy for the reader to see them while reading the manuscript). Also: the justification of the choice of the mutant sequence (nt 13-20 being AAGUGUCC instead of UCCAUGGG) is very obscure (what is an "anti-complementary sequence"? I know what a "complementary sequence" is, I know what an "antisense sequence" is - and these are the same thing; but an "anti-complementary sequence" is very mysterious to me; at least I can see that it is not the complementary sequence of the wild-type 13-20 subsequence). Please detail.
4. Page 6, sentence "Absolute expression of the mature [...]": Fig. 1H shows some large differences in wt vs. mutant miRNA abundance in at least two samples (kidney and CD8+), so the text should read "in most tissues and cell populations examined", not "all tissues and cell populations".
5. Page 7, sentence "However, none of these phenotypes [...]": please define "aged" mice (the next sentence clearly defines what an "extremely advanced age" is: please do the same here).
6. Page 10, sentence "Because RNA-interference in mammals can mediate [...]". I know there is sometimes much confusion about that term in the literature, but note that "RNA interference" is different from the "miRNA pathway". RNA interference is a sequence-specific repression triggered by a double-stranded RNA (which is converted into siRNAs by Dicer).
7. Page 22, sentence "For comparisons between two groups, [...]": is the assumption of equal variances always justified throughout the manuscript? (in other words: have the authors checked for variance homogeneity?)
8. Page 28, Fig. 1E: maybe it's just an image resolution issue, but: I can't see a WT band for the "miR-146a +/3' F lane" (second from last), while I was expecting one. Anything wrong with that sample?
9. Page 28, Fig. 1D: how can we be sure that the band shown with an asterisk is really due to cross-hybridization? If there is no argument supporting it, it should be flagged as "probable" cross-

hybridizing band.

10. Page 33, Fig. 5E and 5F: in Fig. 5F, the y-axis title reads "Relative IRAK1". Relative to what? Multiple normalization procedures can be imagined here (relative to "-LPS" or not; relative to GAPDH or not), and the legend does not help much. Understanding how exactly the data has been processed would help the reader understand why panel F seems to contradict panel E (which shows clear differences in "-LPS" signals across genotypes). Also, please explain why there are two lanes in each experimental condition for Fig. 5E (two lanes for "WT -LPS", two lanes for "WT +LPS", ...): are these biological duplicates? Then the s.e.m. shown in Fig. 1F would probably be much larger. I am a bit puzzled by these two panels.

11. Page 36, Fig. 7: what are the "Controls" shown in the rightmost region of both panels (A and B)? They come after a list of gene names, so I guess these are control UTRs, but then why would they respond to the wt miRNA? Please improve the legend.

Reviewer #2 (Comments to the Authors (Required)):

The manuscript by Bertolet et al, studies the role of miRNA 3' pairing in vivo in a mammalian system. In fact, although the seed sequence of a miRNA is usually sufficient and necessary for silencing, some recent studies have proposed a role for the sequence beyond the seed in mammalian cells and *C. elegans*.

To investigate the role of 3' pairing in vivo in mammalian system, the authors studied miR-146a, whose deletion was already shown to induce several immune defects.

Specifically, they generated a mouse harboring a mutation in the 3' sequence of miR-146a (miR-146a 3'F). They reasoned that, if such 3' sequence is necessary for the miRNA activity, then mice harboring such mutation (3'F mice) should show altered target repression and, possibly, a phenotype.

After careful phenotypic analysis, the 3'F mutants did not show phenotypic alterations and instead resembled wild type mice. This was in contrast to miR-146a KO mice, which show strong immune defects (such as splenomegaly and LPS hypersensitivity). Therefore, the authors conclude that the 3' pairing of such miRNA is dispensable.

Major Points:

The comprehensive phenotypic analysis, which reveals wild type appearance of the 3'F mice, strongly suggests that miR-146a represses its targets only through the seed sequence. However, a convincing connection between the miRNA loss (already in the KO) and the observed phenotype is missing, thus I am not fully convinced that the results presented are conclusive.

In fact, although the miR-146a KO mice show a strong phenotype, their pathology cannot be definitely connected to loss of miRNA activity (ie. via target mRNA de-repression). For the miRNA to be convincingly connected to the immune defects described in miR-146a KO mice here and in (Boldin, 2011), a rescue experiment (suppression of phenotype by exogenously providing miR-146a in KO mice), analysis of mutant mice carrying a miR-146a with scrambled seed sequence (which should show similar pathology as KO), or of mice carrying mutations in the putative miR-146a targets would have been helpful. For these reasons, I fear that this miRNA is not the best model for the hypothesis (importance of 3'pairing) being tested.

1. There are no global changes in transcripts and proteins in miR-146a 3'F. However, there are no changes in the KO mice either, and this was already suggested in the previous publication describing miR-146a KO mice (Boldin 2011).

I agree that the 3'F mice seem similar to wild type and thus the 3' pairing might be dispensable, but how can we mechanistically connect the altered 3' pairing to miRNA activity if there are no obvious direct targets to study?

2. If we embrace the hypothesis that miR-146a works only through few targets (irak1?), thus explaining the lack of global change in transcripts in KO mice (discussed above), then I would want to see a functional validation of such targets (e.g. miRNA binding site manipulations). Presence of seed match is not direct evidence of miRNA-dependent repression.

3. Boldin et al, had described irak1 and traf6 as bona fide target, whose protein level was affected by miR-146a deletion. However, Bertolet and colleagues cannot reproduce the traf6 misregulation. Could the authors comment on this discrepancy?

4. Experiments carried out in vitro (Suppl. Fig.2) show that the 3'pairing has a function in target repression, as acknowledged by the authors in the first result paragraph. What is the authors' opinion about the fact that the same behavior is not observed in vivo?

Minor notes:

1. The same histological section corresponding to the 3'F mouse is present in three different panels: Fig2J (the same image is shown as both liver and kidney) and in Fig.3D, which describes older mice.

2. Brancati & Grosshans, NAR, 2018 have recently shown additional evidence for 3'pairing function in *C. elegans*.

3. I would move the results shown in S2 to the main text and move Fig.4 to Suppl.

4. Figure 1A is mislabeled. (What is the control?)

5. Legend Fig.2K : n (3'F) = 4-5 ?

6. Details about the blot in Fig.5E are missing; I assume the authors are showing two mice per genotype.

7. Fig. 7: the authors should comment on which kind of control they used for the luc assay.

8. Fig. 7: Do the targets have a seed match only or also 3'pairing? You would not expect any change in targets with a seed match between WT and 3'F mice. If a change has to be seen, it would be in targets with 3'pairing to the wt miR-146a.

9. Consider putting together S1A and S2B-C and S1B-C with S2D-E for immediate visualization of miRNA/target duplex in each luc assay. Please add stats on the plots.

10. In general, please add stats to all figures and plots where appropriate and make homogeneous axis labels (e.g. S3).

Reviewer #3 (Comments to the Authors (Required)):

Review of MS. LSA-2018-00249

In the manuscript entitled: 'MiR-146a wild-type 3' sequence identity is dispensable for proper innate immune function in vivo' the authors perform experiments with newly created miR-146a knockout mice that for the most part recapitulate phenotypes of the immune system that have previously been published. Additionally, miR-146a^{3'F} mice were generated, designed to reduce 3' sequence complementarity with target mRNAs, while at the same time leaving the miR-146a seed sequence unaltered. Extensive and carefully conducted experiments show that these mice in some instances are mildly hypomorphic, but in most aspects are indistinguishable from the wild-type phenotype.

Major points:

1. The authors neglect to mention that miR-146a and miR-146b share the same seed sequence and only differ by two nucleotides in their 3' sequences. At no point do they mention that miR-146b could potentially replace the function of miR-146a^{3'F}. Mild hypomorphic phenotypes might reflect a

mixture of miR-146b and miR-146a3'F activity. For unknown reasons, the complete lack of miR-146a in miR-146a-deficient mice might be harder to compensate by miR-146b. In general I wonder why the authors chose miR-146a over for instance miR-155, which is not part of a miRNA family. The authors should measure and state the miR-146b levels in Figures 1 G-I and 6 A. They need to discuss the possibility of miR-146b compensating for miR-146a3'F and (in Figure 6 A-B) miR-146a function.

2. Although I value the large body of negative data proving that the 3' part of miR-146a does not impact on mouse phenotypes, I don't think the analysis of mRNA and protein expression -- as it is -- is very useful to the readers. It seems that the phenotypic difference between WT, miR-146a3'F compared to KO is not reflected in the transcriptome or proteome of BMDM cells, and either the choice of cells confounds to demonstrate it or other miRNAs or RNA-binding proteins compensate in these cells and under these culture conditions. To improve the presentation the isolation and profiling of myeloid cells from the animals without in vitro culture should be performed to demonstrate valid differences or, alternatively, the nature of compensation should be demonstrated by anti-Ago-IP/Immunoprecipitation or even CLIP-Seq technology.

Minor points:

Suppl. Fig. 2 is presented before Suppl. Fig. 1 in the text. All panels showing alignments should be presented in Suppl. Fig. 1 to avoid this and to make it easier for the reader.

Experiments of Suppl. Fig. 2 would be more convincing if miR-146a and miR-146a3'F had to undergo intracellular processing as pri- or pre-miRNAs.

There are few typos (for example page 8, Results mir-146a3'/- should read miR-1463'F/- or page 11, Discussion, line 4: 'miR-46a' should read 'miR-146a').

We are grateful to the referees for their time, their thoughtful and generally positive and/or constructive reviews of the manuscript, and the issues that they raise to help us improve our final product. Please find point-by-point responses to these issues below.

Reviewer #1 (Comments to the Authors (Required)):

In this manuscript, G. Bertolet and colleagues investigate the functional consequences of mutating the 3'-most sequence of miR-146a, without changing the seed nor the central region. Because conflicting data has been published regarding the contribution of non-seed regions, it is indeed important to probe the consequences of such a mutation (especially when the tested phenotypes include *in vivo*, integrated readouts, as is the case here). The authors find that the mutation triggers some consequences at the microscopic level (when the readout is a target RNA or protein abundance), but they did not find any convincing proof for macroscopic consequences. This is an interesting manuscript, addressing an important question in the field of miRNA-mediated gene regulation. The detailed phenotypic assessment (including many physiological assays, interrogating both homozygous and hemizygous mutants) is exemplary. Several problems (two major points, which seem to affect the authors' interpretations; but also many minor points, which make the manuscript obscure or inconsistent on a few important issues) tend to weaken its conclusions, so I recommend correcting all these points. Once this is done, the manuscript will be a strong candidate for publication in Life Science Alliance.

Major points:

1. Page 5, sentence "Transfection of this vector into HEK 293 cells [...]": to me, Fig. 1A does not support such a conclusion. It is not exactly clear to me what the authors mean with "similarly processed". Do they imply that the mutant miRNA is matured from Drosha and Dicer cleavages occurring at the same positions? If yes, that poorly-resolved Northern blot, showing smeary bands on two distinct membranes (none of them containing a size marker), does not provide enough information on miRNA length to conclude anything (not even mentioning the fact that miRNAs can have the same length but not the same 5' and 3' ends). Or do the authors mean that the mutant miRNA accumulates at the same level than the wt miRNA? If yes, these two membranes, hybridized with two different probes, are also useless for such a conclusion. It is actually possible to use Northern blot for such an assessment, but then each membrane has to be calibrated with synthetic RNA oligos in a few additional lanes, in order to convert Northern blot intensity signals into fmol of RNA (apparently Fig. 1G used calibrating synthetic oligos: these very same oligos could be used to make a Northern blot really quantitative for Fig. 1A).

We agree with the reviewer that the presentation of this figure could be improved, and that we could have certainly used a higher percentage gel for the experiment. We further agree that the experiment does not formally demonstrate that the pre-miRNA was processed with the appropriate "register" that would produce our predicted mutant miRNA. We have replaced the previous figure with another and better labeled iteration of this experiment (including the location of size standards). For the benefit of the reviewer, we note that these data do indeed come from a quantitative Northern blot with spotted synthetic standards, and that the wild-type and mutant

mature miRNA species were produced at 39.5 and 40.1 fmol/ 30 µg total RNA from the transfected in this experiment. Nonetheless, we have softened our wording to reflect the uncertainty in regard to sequence identity at this point in the narrative, and now include small RNA-seq data Supplemental Table 1, Supplemental Figure 3, and supporting source data files) that confirms that the predicted mutant miRNA is produced. These latter data are mentioned at the top of page 7 and further explained in detail below, in response to a concern raised by Reviewer #3.

2. Page 28: Fig. 1H compares WT to 3'F in the left panel, and WT to KO in the right panel. Why are the WT histograms so different between the two panels? This is not just a normalization issue, because even the ratios between samples are clearly different (e.g., compare thymus to spleen in both panels). It may be that the PCR-based quantification is not that precise (that would be perfectly understandable), or that there is a lot of biological variability between the batches of mice. But then, why are the s.e.m. so narrow? Whether there is a lot of technical noise, or a lot of biological variability, this should also reflect in the s.e.m. error bars... In the end, if WT replicates are so poorly reproducible, then the whole assay may not be able to detect efficiently any potential difference between WT and 3'F miRNAs.

*Given the amount of labor that was required to generate each of these subpanels, the comparisons in the experiments depicted in Figures 1H and 1I were run on different mice, on different days, and on different freshly prepared quantitation standards. As such, the disparities are likely derived from both biological and technical variability (for example partial degradation of the standard in a freeze/thaw cycle – Figure 1I was generated after Figure 1H). We respectfully submit that differences in absolute expression are fundamentally different from differences in relative expression, and that given that most of our measurements/comparisons within and between these two panels here are within two-fold (barring cells differentiated in vitro, which might be expected to have a higher degree of absolute variability between different trials given the additional manipulations), that is about as well as can be done using the methodology we have chosen. We note that we explicitly do **not** conclude or state that the expression is identical within the manuscript, explicitly choosing to use the word “similar”. We also note that repeating these experiments via Northern analysis would preclude the inclusion of canonical immune cell classes. We have also softened our assertion from “all” to “most” in this section as requested by this reviewer’s minor point #4.*

Minor points:

1. Page 2, sentence "The first unbiased identification of bona fide miRNA/mRNA target pairs [...]": I would not agree that the results shown in Chi et al. (2009) "largely corroborated" computational predictions. The Chi et al. paper is certainly not very precise regarding the number and identity of CLIP-identified miRNA targets, which does not ease comparison to computational predictions - but at least their number of targets per miRNA, and the high percentage of seedless targets they found, seem to argue against such an optimistic affirmation. Please justify that sentence.

In response to the reviewer's comment, we have replaced the word "largely" with "to some extent." We have also elaborated upon this point to provide specific examples of how we feel the combined work of these authors supported specific predictions of the 'seed-centric' paradigm.

2. Page 3, sentence "This notion would partially explain the strict evolutionary conservation [...]". Another phenomenon seems to explain equally well the conservation of non-seed sequences: because miRNA*s seem to be under a selective pressure to repress seed-matched transcripts (see <https://www.ncbi.nlm.nih.gov/pubmed/21177881>). As the pre-miRNA secondary structure (hairpin-shaped) is also conserved, then the miRNA sequence facing the miRNA* seed has to be conserved - implying that the miRNA 3' end should be conserved even if it did not play any function in miRNA target recognition.

This is indeed a tantalizing and plausible possibility. However, our goal in this study was not to explain the evolutionary conservation of non-seed sequences per se, but rather, to evaluate one specific hypothesis (i.e. miRNA target recognition and/or function) as to why such conservation exists. As such, we feel that bringing up an alternative hypothesis - one which none of our experiments even approach addressing - would be distracting to our readers.

3. Page 4: for the sake of clarity, the sequence of wild-type and 3'F miR-146a should be given in a main figure (they are currently given in supplementary figure 2, which does not make it easy for the reader to see them while reading the manuscript).

The sequence of both alleles was indeed present in Figure 1B. However, the font we used was admittedly small, and so we can see how it could have easily been missed or unreadable. As such, we have enlarged the font. Thank you for pointing this out.

Also: the justification of the choice of the mutant sequence (nt 13-20 being AAGUGUCC instead of UCCAUGGG) is very obscure (what is an "anti-complementary sequence"? I know what a "complementary sequence" is, I know what an "antisense sequence" is - and these are the same thing; but an "anti-complementary sequence" is very mysterious to me; at least I can see that it is not the complementary sequence of the wild-type 13-20 subsequence). Please detail.

We apologize for the less than ideal choice of words on our part. This has been rectified.

4. Page 6, sentence "Absolute expression of the mature [...]": Fig. 1H shows some large differences in wt vs. mutant miRNA abundance in at least two samples (kidney and CD8+), so the text should read "in most tissues and cell populations examined", not "all tissues and cell populations".

We have made this change.

5. Page 7, sentence "However, none of these phenotypes [...]": please define "aged" mice (the next sentence clearly defines what an "extremely advanced age" is: please do the same here).

This is a good point. We have taken out the adjective "aged" and replaced it with "age-matched" here.

6. Page 10, sentence "Because RNA-interference in mammals can mediate [...]". I know there is sometimes much confusion about that term in the literature, but note that "RNA interference" is different from the "miRNA pathway". RNA interference is a sequence-specific repression triggered by a double-stranded RNA (which is converted into siRNAs by Dicer).

That's embarrassing. Thank you for catching that.

7. Page 22, sentence "For comparisons between two groups, [...]": is the assumption of equal variances always justified throughout the manuscript? (in other words: have the authors checked for variance homogeneity?)

The statistical textbook in use by our graduate program, as well as the course director for the associate course (an accomplished professional statistician) both say it is not wise to run an ANOVA to check for equal variances before each and every t-test as a general rule, given that such a practice increases the chance of a Type 1 Error. In response to the reviewer's concern we have run some ANOVAs where appropriate in the manuscript (for example Figures 2C, F-I, and K) and it does indeed appear that our assumption of equal variances is not always true. However, in these cases where it was not true, assessing significance with a Welch's t-test does not impact whether these results are statistically significant or not.

8. Page 28, Fig. 1E: maybe it's just an image resolution issue, but: I can't see a WT band for the "miR-146a +/3'F lane" (second from last), while I was expecting one. Anything wrong with that sample?

The band is clearly visible to us in the submitted files. However, we have altered the contrast if this image in the hope that the band will be more clearly visible. Thank you for pointing out the difficulty some people might have had in seeing this.

9. Page 28, Fig. 1D: how can we be sure that the band shown with an asterisk is really due to cross-hybridization? If there is no argument supporting it, it should be flagged as "probable" cross-hybridizing band.

The argument is that, given that the probe lies entirely within the 8 kb fragment that will result from EcoRV/BamHI double digestion, that 8 kb fragment is clearly present on the gel (the same rationale exists for the 5.1 kb fragment resulting from introduction of the EcoRV site from the targeting vector – the probe does not span this junction), by definition the ~5.4 kb band that is present must represent hybridization of the probe to a genomic fragment from elsewhere in the genome. This argument is strengthened given that (i) the band in question was present in all ES

cell clones examined in this Southern Blot, arguing strongly against non-targeted insertion of a the targeting vector as a transgene, which would be expected to occur randomly, (ii) partial digestion of any of the predicted fragments that we are assaying for is not predicted to yield this ~5.4 kb band, and (iii) the mobility of the band is not affected by the homologous recombination.

10. Page 33, Fig. 5E and 5F: in Fig. 5F, the y-axis title reads "Relative IRAK1". Relative to what? Multiple normalization procedures can be imagined here (relative to "-LPS" or not; relative to GAPDH or not), and the legend does not help much. Understanding how exactly the data has been processed would help the reader understand why panel F seems to contradict panel E (which shows clear differences in "-LPS" signals across genotypes). Also, please explain why there are two lanes in each experimental condition for Fig. 5E (two lanes for "WT - LPS", two lanes for "WT +LPS", ...): are these biological duplicates? Then the s.e.m. shown in Fig. 1F would probably be much larger. I am a bit puzzled by these two panels.

We have clarified our explanation of how these data were analyzed.

11. Page 36, Fig. 7: what are the "Controls" shown in the rightmost region of both panels (A and B)? They come after a list of gene names, so I guess these are control UTRs, but then why would they respond to the wt miRNA? Please improve the legend.

Thank you for pointing this out. It was indeed confusing. The controls are the very same constructs used in what is now Supplemental Fig S1, and this is now noted in the revised figure legend for this figure.

Reviewer #2 (Comments to the Authors (Required)):

The manuscript by Bertolet et al, studies the role of miRNA 3' pairing in vivo in a mammalian system. In fact, although the seed sequence of a miRNA is usually sufficient and necessary for silencing, some recent studies have proposed a role for the sequence beyond the seed in mammalian cells and *C. elegans*.

To investigate the role of 3' pairing in vivo in mammalian system, the authors studied miR-146a, whose deletion was already shown to induce several immune defects.

Specifically, they generated a mouse harboring a mutation in the 3' sequence of miR-146a (miR-146a 3'F). They reasoned that, if such 3' sequence is necessary for the miRNA activity, then mice harboring such mutation (3'F mice) should show altered target repression and, possibly, a phenotype.

After careful phenotypic analysis, the 3'F mutants did not show phenotypic alterations and instead resembled wild type mice. This was in contrast to miR-146a KO mice, which show strong immune defects (such as splenomegaly and LPS hypersensitivity). Therefore, the authors conclude that the 3' pairing of such miRNA is dispensable.

Major Points:

The comprehensive phenotypic analysis, which reveals wild type appearance of the 3'F mice, strongly suggests that miR-146a represses its targets only through the seed sequence. However, a convincing connection between the miRNA loss (already in the KO) and the observed phenotype is missing, thus I am not fully convinced that the results presented are conclusive.

In fact, although the miR-146a KO mice show a strong phenotype, their pathology cannot be definitely connected to loss of miRNA activity (ie. via target mRNA de-repression). For the miRNA to be convincingly connected to the immune defects described in miR-146a KO mice here and in (Boldin, 2011), a rescue experiment (suppression of phenotype by exogenously providing miR-146a in KO mice), analysis of mutant mice carrying a miR-146a with scrambled seed sequence (which should show similar pathology as KO), or of mice carrying mutations in the putative miR-146a targets would have been helpful. For these reasons, I fear that this miRNA is not the best model for the hypothesis (importance of 3'pairing) being tested.

With all of our respect, we are confused by the reviewer's line of logic. Any pathology/phenotype observed within the context of a defined genetic lesion is, by convention, defined as a product of that genetic lesion. This is the case whether in an animal or in an isolated cell type, both of which we have examined here. The rescue experiments proposed are done in this context with alleles – both the wild-type or the 3'F allele demonstrably offset the phenotype and pathology associated with miR-146a loss-of-function. Indeed, this offset can be assumed to be a product of correct temporal and spatial expression, and expression at physiological levels given that the rescue is provided as a “knock-in” to the endogenous locus. It is unclear to us what the benefits of providing miR-146a in another fashion might be. While we agree that an allele of miR-146a with a scrambled seed would be interesting, as would germline manipulations of established miR-146a targets, such experiments are not trivial and are beyond the scope of this work.

1. There are no global changes in transcripts and proteins in miR-146a 3'F. However, there are no changes in the KO mice either, and this was already suggested in the previous publication describing miR-146a KO mice (Boldin 2011). I agree that the 3'F mice seem similar to wild type and thus the 3' pairing might be dispensable, but how can we mechanistically connect the altered 3' pairing to miRNA activity if there are no obvious direct targets to study?

Our work is an in vivo analysis to determine whether miR-146a 3' sequence identity (and thus – presumably - pairing) is required for this miRNA's genetic function, as defined by the phenotype of mice deficient for this miRNA. Activity (or perhaps function would be a more accurate word here) is then measured using phenotype as a proxy. Of course, the model system that we have chosen precludes several subtler approaches and/or questions that might be asked in an in vitro system, but we do feel that it provides valuable information nonetheless. Beyond this, it is unclear how a “micromanager” mode of regulation, affecting hundreds of transcripts globally, would be experimentally parsed for mechanism. We do however demonstrate differences in regulation of Irak1, which is an (now a better – see below) established miR-146a target with an established role in NFkB activation and cytokine production.

2. If we embrace the hypothesis that miR-146a works only through few targets (irak1?), thus explaining the lack of global change in transcripts in KO mice (discussed above), then I would want to see a functional validation of such targets (e.g. miRNA binding site manipulations). Presence of seed match is not direct evidence of miRNA-dependent repression.

Our hats are off to the reviewer for catching this. These experiments have been performed for human IRAK1 (and TRAF6), and given the preponderance of work that has been published examining the regulatory relationship between miR-146a and these gene products in both mice and human, we were floored that when we went back to find the experiments that established this for the murine 3' UTR we were unable to find anything. It seems that given the human demonstration, evolutionary conservation, and functional relationships that this has been taken for granted to this point by the scientific community.

In any case, we had of course performed these experiments alongside the original dose-responses and omitted the data in the previous submission given the above misperception. These data are now included in Figure S2. We note that generating another allele in which the MREs of Irak1 are mutated or disrupted for in vivo analysis would further support our model and the models of others, but this is certainly not trivial and beyond the scope of the current work.

3. Boldin et al, had described irak1 and traf6 as bona fide target, whose protein level was affected by miR-146a deletion. However, Bertolet and colleagues cannot reproduce the traf6 misregulation. Could the authors comment on this discrepancy?

Interestingly, although heightened expression of TRAF6 was previously reported in Boldin et al., 2011 as correctly pointed out by the reviewer, we did not observe this in any of multiple (n = 7 including steady-state and LPS challenge) in vitro experiments performed in multiple manners. We have no explanation for the discrepancy between our results and those previously published in this system, and can only say that, in our hands, no difference is observed. This, however, is not without precedent in the literature. Traf6 levels have been reported to be seemingly unaffected by miR-146a in LPS-stimulated B cells (Jiang et al., 2018), and in the PBMCs of patients with Sjögren's syndrome (in which miR-146a levels are elevated), Traf6 levels are also abnormally high when compared to healthy controls (Zilahi et al., 2012). This has been added to the discussion.

4. Experiments carried out in vitro (Suppl. Fig.2) show that the 3' pairing has a function in target repression, as acknowledged by the authors in the first result paragraph. What is the authors' opinion about the fact that the same behavior is not observed in vivo?

Most simply put, we feel that the discrepancy between our in vitro and in vivo results is a reflection on the limitation of in vitro reporter/validation systems, and what may be drawn from them. We do note that siRNA "off-target" phenomena and spectra are well-established in the literature, and while the spectrum of putative targets for a given miRNA is computationally defined in part by evolutionary conservation of seed-complementary sequences within these

targets, the spectrum of “off-target” transcripts is defined solely by the presence of an appropriate seed-match and response to the presence of the small RNA species provided, with no requirement for conservation. Ergo, it seems reasonable to infer that responses of certain 3' UTR reporters to an exogenously provided miRNA in vitro may exaggerate a corresponding physiological regulatory relationship. However, given the myriad of other possibilities (e.g. native versus reporter sequence identity, protein association, etc.), the controversy that continues to surround miRNA target recognition and biological function, and what we feel we can confidently draw from our profiling data, we feel it is most wise to limit our speculation on this point within the manuscript itself.

Minor notes:

1. The same histological section corresponding to the 3'F mouse is present in three different panels: Fig2J (the same image is shown as both liver and kidney) and in Fig.3D, which describes older mice.

That is horrifying. Please accept our most sincere apologies and thank you so much for noticing this.

2. Brancati & Grosshans, NAR, 2018 have recently shown additional evidence for 3' pairing function in *C. elegans*.

The reviewers have our sincere thanks for bringing this study, which is now cited in the introduction, to our attention.

3. I would move the results shown in S2 to the main text and move Fig.4 to Suppl.

We can see the merit in this idea, but we hope that the reviewer will recognize the merit of our own existing choice of presentation here.

4. Figure 1A is mislabeled. (What is the control?)

The labeling for Figure 1A was indeed incorrect. Given this, and concerns raised by Reviewer 1, we have replaced the figure with another iteration of the experiment that is more appropriately labeled and altered our statement of the conclusions that can be drawn from the figure. Thank you.

5. Legend Fig.2K : n (3'F) = 4-5 ?

This was inadequately described. 5 for the liver, 4 for the kidney. The figure legend has been updated to reflect this.

6. Details about the blot in Fig.5E are missing; I assume the authors are showing two mice per genotype.

The reviewer is correct. Thank you for pointing out the lack of detail. The figure legend has been updated to rectify this.

7. Fig. 7: the authors should comment on which kind of control they used for the luc assay.

Thank you for catching that. We apologize for the omission.

8. Fig. 7: Do the targets have a seed match only or also 3'pairing? You would not expect any change in targets with a seed match between WT and 3'F mice. If a change has to be seen, it would be in targets with 3'pairing to the wt miR-146a.

The targets within Figure 7 were based on the criteria of (i) having a canonical seed match, and (ii) exhibiting differences in relative expression that would be consistent with a difference in regulation among the three alleles evaluated. Predicted 3' pairing was not a criterium for classification. Nonetheless, alignments of the miRNAs with each of these potential targets are now provided in a new supplementary figure.

9. Consider putting together S1A and S2B-C and S1B-C with S2D-E for immediate visualization of miRNA/target duplex in each luc assay. Please add stats on the plots.

We agree that this would facilitate comprehension of the figures, and they have been altered accordingly.

10. In general, please add stats to all figures and plots where appropriate and make homogeneous axis labels (e.g. S3).

At the reviewer's request, we have carefully surveyed our figures and plots to add any missing statistics and have improved the homogeneity of axis labels in Figure S3.

Reviewer #3 (Comments to the Authors (Required)):

Review of MS. LSA-2018-00249

In the manuscript entitled: 'MiR-146a wild-type 3' sequence identity is dispensable for proper innate immune function in vivo' the authors perform experiments with newly created miR-146a knockout mice that for the most part recapitulate phenotypes of the immune system that have previously been published. Additionally, miR-146a3'F mice were generated, designed to reduce 3' sequence complementarity with target mRNAs, while at the same time leaving the miR-146a seed sequence unaltered. Extensive and carefully conducted experiments show that these mice in some instances are mildly hypomorphic, but in most aspects are indistinguishable from the wild-type phenotype.

Major points:

1. The authors neglect to mention that miR-146a and miR-146b share the same seed sequence and only differ by two nucleotides in their 3' sequences. At no point do they mention that miR-146b could potentially replace the function of miR-146a3'F.

The reviewer is correct, and we have inserted text in the introduction to acknowledge the similarity between miR-146a and miR-146b. In regard to the second point, given that the miR-146a null mice have a robust phenotype, it is clear, by definition, that miR-146b cannot inherently compensate for the function of miR-146a in a physiological setting. This does not exclude the possibility that miR-146b might be able to do this if provided exogenously at levels appropriate to effect this function, but that is not the focus of this work.

Mild hypomorphic phenotypes might reflect a mixture of miR-146b and miR-146a3'F activity. For unknown reasons, the complete lack of miR-146a in miR-146a-deficient mice might be harder to compensate by miR-146b.

While the former statement above is formally true, we present here a genetic study that isolates miR-146a function through deficiency. The miR-146a phenotype is inherently a product of miR-146a deficiency and wild-type miR-146b physiological function, and while it would certainly be interesting to assess whether a compound deficiency in miR-146a and miR-146b exacerbated or altered the phenotype established for miR-146a-deficient mice and cells, the point stands that miR-146b genetic function is intact in both miR-146a deficient mice and miR-146a3'F mice. Thus, from a genetic perspective, we feel that with the data we provide we have adequately demonstrated that the miR-146a3'F almost completely restores miR-146a function, since wild-type miR-146b function is present in both contexts. The reviewer's alternative model would certainly be interesting to test in the context of additional mutant alleles, but we would have to generate these alleles de novo and respectfully submit that this goes beyond the scope of this study. In any case, this issue is now briefly addressed in the discussion.

In general I wonder why the authors chose miR-146a over for instance miR-155, which is not part of a miRNA family.

This is a fair question to ask. We chose miR-146a over miR-155 because of the extensive characterization and fairly robust (for a miRNA) phenotype in the previously established miR-146a deficiency model. Beyond having several "angles to attack" in looking for differences in phenotype between the miR-146a WT and mutant alleles, we anticipated that this model would provide us the best context in which to identify modest differences in genetic function – e.g. mild hypomorphic function.

The authors should measure and state the miR-146b levels in Figures 1 G-I and 6 A.

We have not measured and stated the levels of miR-146b in Figures 1 G-I, under our above-stated rationale that this is a study isolating (and assessing the degree of rescue of) the function of miR-146a by a miR-146a mutant allele and we neither notice phenotypes nor study any of the

relevant tissues there. We do, however, now provide results of small RNA sequencing of a subset of the BMDMs underlying Figures 6A and 6B in the new Supplementary Table 1, Supplementary 3, and these datasets have been deposited. These new data provide two key pieces of information: (i) that miR-146a3'F is produced in mutant cells at the correct register, and (ii) that while miR-146b is increased in expression in miR-146a-deficient cells, perhaps in response to pressure for functional compensation, no concomitant increase in miR-146b expression is observed in miR-146a3'F cells. These data, along with our extensive phenotypic characterization of the 3'F mice, support both the notion that this pressure for functional compensation does not exist in miR-146a3'F cells.

In regard to the new data that we provide, it is also important to rationalize to the reviewers the comparatively low levels of miR-146a3'F that are observed in the short RNA sequencing data, especially given the qRT-PCR data that we provide in Figure 1H. T4 RNA ligase (and its various laboratory-produced and commercially available mutants) differs from DNA ligase in that it does not ligate all products together with the same efficiency. Although this was described by Olke Uhlenbeck and coworkers over 30 years ago, the extent and scale of this characteristic is not broadly recognized, even within the short RNA field. Indeed, the representation of a given short RNA species within a T4 RNA ligase-generated library is a direct function of both the identity of the short RNA and the cloning linker used, and how changes in the identity of the linker or the short RNA to be ligated will impact the relative efficiency of a given reaction is impossible to predict a priori. Thus, while short RNA libraries prepared using a ligation-dependent step may be “horizontally” compared across libraries constructed with the same methodology for relative expression (the relative ligation efficiency for a given short RNA and common linker are consistent across these libraries and thus constant), “vertical” comparisons of relative expression of separate small RNA species within an individual library or “diagonally” comparing the relative abundance of two different species among two or more libraries is often not likely to be accurate. The experience of both myself (the corresponding author) and our collaborators suggests that these differences are unpredictable and can be of significant magnitude. My own experience “bar-coding” small RNA libraries from different genotypes in another project revealed that the resulting libraries would invariably cluster on linkers rather than genotype (even if the differences among the genotypes within the libraries had already been established), and our colleagues’ switching library kits (and thus linkers) in their own work dropped the representation of an established T lymphocyte miRNA that they study from 7000 relative representations per million to zero, even in the context of the latter libraries vigorously passing all relevant wet-bench and computational QC.

Ultimately then, the point that we are trying to make is that because the qRT-PCR methodology used to provide the data in Figure 1H is ligation-independent, and because these data are calculated from a dilution of a synthetic standard in total RNA from *D. melanogaster* (which does not encode miR-146 family members), it is a far more accurate quantitation of absolute expression of miR-146a3'F within the BMDMs than ligation-dependent small RNA sequencing data.

They need to discuss the possibility of miR-146b compensating for miR-146a3'F and (in Figure 6 A-B) miR-146a function.

A discussion of this, following the rationale that we provide above, is now included in the discussion.

2. Although I value the large body of negative data proving that the 3' part of miR-146a does not impact on mouse phenotypes, I don't think the analysis of mRNA and protein expression -- as it is -- is very useful to the readers. It seems that the phenotypic difference between WT, miR-146a3'F compared to KO is not reflected in the transcriptome or proteome of BMDM cells, and either the choice of cells confounds to demonstrate it or other miRNAs or RNA-binding proteins compensate in these cells and under these culture conditions. To improve the presentation the isolation and profiling of myeloid cells from the animals without in vitro culture should be performed to demonstrate valid differences or, alternatively, the nature of compensation should be demonstrated by anti-Ago-IP/Immunoprecipitation or even CLIP-Seq technology.

We respectfully disagree in regard to usefulness of our data to the readers and the field. While we acknowledge that a "shift" in predicted mRNA or protein targets is routinely documented in response to ectopic overexpression of a miRNA and in some cases due to its loss of function, our data do recapitulate previous analysis of miR-146a-deficient BMDMs by the Baltimore group and do so in the context of a cell type with a defined in vitro phenotype due to a defined genetic lesion. Our own qRT-PCR data show that miR-146a is most highly expressed in this cell type, and so one must assume that any phenotype here is due to the deficiency in miR-146a function. So while this was not, perhaps, the expected answer, we are fairly confident in asserting it is indeed the correct one, and we would be hard pressed to come up with an alternate explanation beyond the one that we explicitly show – that IRAK1, an established miR-146a target, is dysregulated in the cells.

Nonetheless, the reviewer's suggestions for alternative methods is well taken, and had the proposed experiments been technically feasible we certainly would have executed them. Isolation of native macrophages in sufficient numbers for these profiling experiments (particularly immunoprecipitations or mass spectrometry) is extremely difficult and would require pooling prohibitively large numbers of age-matched mice. Beyond this, previous discussions between the PI and experts in eCLIP have revealed that a robust and highly controlled/reproducible protocol for Argonautes in this context does not currently exist. Adding to the complexity, to have a definitive answer we would need to be able to simultaneously pull down all four Argonautes for analysis - a Herculean endeavor in and of itself - and be confident that our occupancy data derived from this was quantitative, which is not generally accepted as the case. Such an endeavor is likely beyond the technical capability of our laboratory and, even were it not, we respectfully suggest that the technological development required for this latter approach, even if performed in collaboration, is beyond the scope of this manuscript.

Minor points:

Suppl. Fig. 2 is presented before Suppl. Fig. 1 in the text. All panels showing alignments should be presented in Suppl. Fig. 1 to avoid this and to make it easier for the reader.

We have re-assigned different subfigures between Fig S1 and S2 to facilitate viewing, in keeping with the comments of multiple reviewers.

Experiments of Suppl. Fig. 2 would be more convincing if miR-146a and miR-146a3'F had to undergo intracellular processing as pri- or pre-miRNAs.

The reviewer's point is well taken, and we can see the merit in this notion. Our intention here, however, was to test the function of the mature miRNA in a fashion that exclude any differences that might arise from altered efficiency in precursor processing or loading.

There are few typos (for example page 8, Results mir-146a3'/- should read miR-1463'F/- or page 11, Discussion, line 4: 'miR-46a' should read 'miR-146a').

We have corrected all of these that we have found, including the ones brought to our attention here. Thank you very much.

January 28, 2019

RE: Life Science Alliance Manuscript #LSA-2018-00249R

Dr. Joel R Neilson
Baylor College of Medicine
Molecular Physiology and Biophysics
One Bayor Plaza, MS BCM 335
Room S444
Houston, TX 77030

Dear Dr. Neilson,

Thank you for submitting your revised manuscript entitled "MiR-146a wild-type 3' sequence identity is dispensable for proper innate immune function in vivo". As you will see, reviewer #1 re-evaluated your work again and now supports publication of a slightly further revised version of your manuscript in Life Science Alliance. I would thus like to invite you to address the minor comments made by the reviewer and to submit a final version to us. Please also address the following points when revising your work:

- please add a callout to figure panel 2K, S2E and S2F in the manuscript text
- please add a label for figure panel 3H in figure 3
- please add a 'C' to the figure legend for SFig5
- please add error bar descriptions in all figure legends, a few are currently missing (eg. for panels in Fig 1, 5, 7)

A. FINAL FILES:

-- High-resolution figure, supplementary figure and video files uploaded as individual files: See our detailed guidelines for preparing your production-ready images, <http://life-science-alliance.org/authorguide>

-- Summary blurb (enter in submission system): A short text summarizing in a single sentence the study (max. 200 characters including spaces). This text is used in conjunction with the titles of papers, hence should be informative and complementary to the title. It should describe the context

and significance of the findings for a general readership; it should be written in the present tense and refer to the work in the third person. Author names should not be mentioned.

B. MANUSCRIPT ORGANIZATION AND FORMATTING:

Full guidelines are available on our Instructions for Authors page, <http://life-science-alliance.org/authorguide>

Thank you for your attention to these final processing requirements.

Sincerely,

Reviewer #1 (Comments to the Authors (Required)):

This revised version addressed my concerns (as well as those from the other two referees) in a

satisfying manner, and I now support publication, provided that the few remaining minor points are corrected:

Minor points:

1. Supplemental Table S1 has apparently been cropped. The legend is hardly readable (the end of each line is missing), and the second part of the table (showing mutant data) is just non-usable (miRNA names are missing, the reader cannot even find the data for miR-146a or for the 3F mutant miRNA). Also, that table only shows read abundances for official miRNA sequences, but it does not show the location of the actual 5' and 3' ends of the detected miRNA reads (while the manuscript says that the mutant miRNA "was processed at the correct register"). The text actually reads "Supplemental Table S1 and associated primary dataset", as if the reader was expected to fetch the raw data from SRA, then map it on pre-miRNA hairpins, and analyze the statistics of Dicer and Drosha cleavage site usage... This is not acceptable: please provide a figure (e.g., a Supplemental Figure) showing the location of observed 5' and 3' ends for wt and 3' F miR-146a (for example: a plot showing nt coordinate along the hairpin, on the x-axis, as well as the number of detected reads whose 5' end or whose 3' end maps on that nucleotide, on the y-axis; the wt and mutant miRNA curves will make the comparison straightforward).
2. See minor point #2 in my original review: I insist, it is important to mention that alternative possibility. Otherwise, the reader will be tempted to see this conservation signal as a proof of miRNA 3' end functionality.
3. See minor point #3 in my original review: thank you, now I understand what has been done (basically, each nucleotide was replaced by the nucleotide which is facing it in the pre-miRNA precursor). But I find the new formulation (on page 4) still misleading (it says that each nucleotide was replaced by the "corresponding paired nucleotide", but some of them are not paired: there's an A-C mismatch). And the next sentence adds even more confusion by saying that the 3' region of the mutant miRNA is set to be "complementary to its native sequence", which it clearly is not. The complementary sequence would be the reverse-complement of the native sequence; here, the mutant sequence is almost the complement (not exactly the complement, because of G-U wobbles and because of the A-C mismatch), but not the "reverse" of the native sequence... Please clarify this in the text (e.g., "each nucleotide was replaced by the nucleotide which is facing it in the pre-miRNA precursor"). For the same reason, the abstract should be corrected ("the 3' half has been altered to be complementary to the wild-type sequence" is incorrect), as well as the Introduction (page 3: "Within this allele, the sequence of the 3' half [...]").
4. See minor point #7 in my original review: I am a bit puzzled by the authors' response. When I was asking whether variance homogeneity had been verified, I was not asking for an ANOVA test (which is something completely different! even though it is called "analysis of variance", the ANOVA is in fact not a way to test the similarity of variances across samples, but the similarity of means across samples). Variance homogeneity is most classically assessed by Levene's test (alternatives exist, and they are certainly acceptable - but ANOVA is not one of them).
5. (sorry for having missed that one in my original review) The definition of the "seed" has always been a bit fuzzy, with some authors defining it as nt 2-7 while others consider it is nt 2-8. In his last review (<https://www.ncbi.nlm.nih.gov/pubmed/29570994>), D. Bartel proposes to name 2-7 the "seed" and 2-8 the "seed region". This is certainly not the least ambiguous nomenclature, but at least it distinguishes the two things. If the authors want to use the "2-8" definition, I thus recommend calling it the "seed region" instead of the "seed".

We are grateful to both the editorial staff and Reviewer 1 for alerting us to additional omissions and issues to be remedied in the manuscript. Please find our point-by-point responses to these below:

Editorial Requests:

- please add a callout to figure panel 2K, S2E and S2F in the manuscript text

Done.

- please add a label for figure panel 3H in figure 3

Done.

- please add a 'C' to the figure legend for SFig5

Done.

- please add error bar descriptions in all figure legends, a few are currently missing (eg. for panels in Fig 1, 5, 7)

Done. Thank you so much for alerting us to these issues.

Reviewer 1 Additional Requests:

1. Supplemental Table S1 has apparently been cropped. The legend is hardly readable (the end of each line is missing), and the second part of the table (showing mutant data) is just non-usable (miRNA names are missing, the reader cannot even find the data for miR-146a or for the 3F mutant miRNA). Also, that table only shows read abundances for official miRNA sequences, but it does not show the location of the actual 5' and 3' ends of the detected miRNA reads (while the manuscript says that the mutant miRNA "was processed at the correct register"). The text actually reads "Supplemental Table S1 and associated primary dataset", as if the reader was expected to fetch the raw data from SRA, then map it on pre-miRNA hairpins, and analyze the statistics of Dicer and Drosha cleavage site usage... This is not acceptable: please provide a figure (e.g., a Supplemental Figure) showing the location of observed 5' and 3' ends for wt and 3' F miR-146a (for example: a plot showing nt coordinate along the hairpin, on the x-axis, as well as the number of detected reads whose 5' end or whose 3' end maps on that nucleotide, on the y-axis; the wt and mutant miRNA curves will make the comparison straightforward).

This is of course natively an Excel file, and it would appear that the manuscript submission portal converted this to a PDF. Had we known that the reviewer would not have access to the Excel spreadsheet, we certainly would have taken more care in the page setup formatting for this file. Sorry. In addition, we ultimately agree with the reviewer that inclusion of the requested graphs would provide an additional line of evidence in evaluating our allele and our work. This is now included in Figure S3.

2. See minor point #2 in my original review: I insist, it is important to mention that alternative possibility. Otherwise, the reader will be tempted to see this conservation signal as a proof of miRNA 3' end functionality.

The point of this paragraph is that the 3' end of the miRNA has been recently shown by several groups to play a role in target recognition and function. Our initial statement in regard to evolutionary conservation was a bit tangential to the main point, and while the reviewer's concern is acknowledged, inclusion of this alternate possibility in addition to our already tangential statement will significantly disrupt the logic flow within this section of the introduction. As a compromise, we have deleted the original statement.

3. See minor point #3 in my original review: thank you, now I understand what has been done (basically, each nucleotide was replaced by the nucleotide which is facing it in the pre-miRNA precursor). But I find the new formulation (on page 4) still misleading (it says that each nucleotide was replaced by the "corresponding paired nucleotide", but some of them are not paired: there's an A-C mismatch). And the next sentence adds even more confusion by saying that the 3' region of the mutant miRNA is set to be "complementary to its native sequence", which it clearly is not. The complementary sequence would be the reverse-complement of the native sequence; here, the mutant sequence is almost the complement (not exactly the complement, because of G-U wobbles and because of the A-C mismatch), but not the "reverse" of the native sequence... Please clarify this in the text (e.g., "each nucleotide was replaced by the nucleotide which is facing it in the pre-miRNA precursor"). For the same reason, the

abstract should be corrected ("the 3' half has been altered to be complementary to the wild-type sequence" is incorrect), as well as the Introduction (page 3: "Within this allele, the sequence of the 3' half [...]").

Thank you for your help in allowing us to state this in the most accurate way in the text. We agree that this representation is superior to the one that we had previously used. This has been rectified.

4. See minor point #7 in my original review: I am a bit puzzled by the authors' response. When I was asking whether variance homogeneity had been verified, I was not asking for an ANOVA test (which is something completely different! even though it is called "analysis of variance", the ANOVA is in fact not a way to test the similarity of variances across samples, but the similarity of means across samples). Variance homogeneity is most classically assessed by Levene's test (alternatives exist, and they are certainly acceptable - but ANOVA is not one of them).

We regret our misunderstanding of the reviewer's original intent, yet we feel that this is a bit of a subjective issue. Tests of variance, within the context of a standard "wet-bench" experiment, are notoriously underpowered for their intended purpose because the n's that are required to formally reject the null hypothesis that there is homogeneity in the variances being compared far exceed the n's collected by convention within this context. This means these tests, in this context, cannot be considered reliable. Invoking the central limit theorem, we could thus argue that any non-homogeneity in variance between or among any of our comparisons is not due to an inherent variance homogeneity within the process itself as a function of genotype but instead a function of the limited n of measurements for each group, and that the variances would approach homogeneity as this n increased. In any case, we ran through each of our t-tests once again to calculate significance (or lack thereof) under the assumption of unequal variance. While of course this resulted in slightly different p values, it did not alter the level of significance indicated within the figures or figure legends in either case. We've now indicated this in the Materials and Methods section as follows:

For comparisons between two groups, statistical significance was assessed using a two-tailed Student's t-test both with and without assumption of equal variance. Neither assumption impacted the level (or lack thereof) of significance depicted in any representation.

5. (sorry for having missed that one in my original review) The definition of the "seed" has always been a bit fuzzy, with some authors defining it as nt 2-7 while others consider it is nt 2-8. In his last review (<https://www.ncbi.nlm.nih.gov/pubmed/29570994>), D. Bartel proposes to name 2-7 the "seed" and 2-8 the "seed region". This is certainly not the least ambiguous nomenclature, but at least it distinguishes the two things. If the authors want to use the "2-8" definition, I thus recommend calling it the "seed region" instead of the "seed".

We were significantly impressed by the level of detail and attention that went into the original review, so no apology on this point is necessary. It's another good point. We've incorporated this change where we deemed appropriate in the manuscript. Thank you for all of these comments.

February 1, 2019

RE: Life Science Alliance Manuscript #LSA-2018-00249RR

Dr. Joel R Neilson
Baylor College of Medicine
Molecular Physiology and Biophysics
One Bayor Plaza, MS BCM 335
Room S444
Houston, TX 77030

Dear Dr. Neilson,

Thank you for submitting your Research Article entitled "MiR-146a wild-type 3' sequence identity is dispensable for proper innate immune function in vivo". I appreciate the introduced changes and it is a pleasure to let you know that your manuscript is now accepted for publication in Life Science Alliance. Congratulations on this interesting work.

DISTRIBUTION OF MATERIALS:

Again, congratulations on a very nice paper. I hope you found the review process to be constructive and are pleased with how the manuscript was handled editorially. We look forward to future exciting submissions from your lab.

Sincerely,

Andrea Leibfried, PhD
Executive Editor
Life Science Alliance
Meyerohofstr. 1
69117 Heidelberg, Germany
t +49 6221 8891 502
e a.leibfried@life-science-alliance.org
www.life-science-alliance.org